# Margin-Adaptive Confidence Ranking for Reliable LLM Judgement

**Gaojie Jin** [1]   **Yong Tao** [1]   **Lijia Yu** [2]   **Tianjin Huang** [1][3]

## Abstract

Jung et al. (2025) introduce a hypothesis testing framework for guaranteeing agreement between large language models (LLMs) and human judgments, relying on the assumption that the model's estimated confidence is monotonic with respect to human-disagreement risk. In practice, however, this assumption may be violated, and the generalization behavior of the confidence estimator is not explicitly analyzed. We mitigate these issues by learning a dedicated confidence estimator instead of relying on heuristic confidence signals. Our approach leverages simulated annotator diversity and a margin-based ranking formulation to explicitly model how confidently an LLM distinguishes between human-agreement and human-disagreement cases. We further derive generalization guarantees for this estimator, revealing a margin-dependent trade-off that informs the design of an adaptive estimator training procedure. When integrated into fixed-sequence testing, the learned confidence estimator yields improved ranking accuracy and empirically strengthens the monotonic relationship between confidence and disagreement risk, leading to higher success rates in satisfying target agreement levels across multiple datasets and judge models.

## 1. Introduction

Large language models are increasingly used as evaluators to judge output quality and preference alignment (Zheng et al., 2023; Dubois et al., 2023; Park et al., 2025; Chiang & Lee, 2023). While this offers scalable, low-cost alternatives to human annotation, a fundamental challenge remains (Xiong et al., 2024): how can we make LLM-as-

[1]Department of Computer Science, University of Exeter [2]Institute of AI for Industries, Chinese Academy of Sciences [3]Department of Mathematics and Computer Science, Eindhoven University of Technology. Correspondence to: Tianjin Huang <t.huang2@exeter.ac.uk>.

*Proceedings of the 43${}^{rd}$ International Conference on Machine Learning*, Seoul, South Korea. PMLR 306, 2026. Copyright 2026 by the author(s).

a-judge decisions reliably trustworthy for downstream use, particularly when the judge reports high confidence?

Recent principled methods have been proposed to improve the reliability of LLM-as-a-judge systems. For instance, Yadkori et al. (2024) introduce conformal abstention to better align LLM judgments with human evaluations, while Mohri & Hashimoto (2024) use conformal prediction to provide high-probability correctness guarantees. Building on these ideas, Jung et al. (2025) develop an unsupervised confidence estimator and derive an exact upper bound on disagreement risk conditioned on a calibration set.

A key assumption underlying such confidence-thresholding procedures is monotonicity: instances with higher estimated confidence should correspond to lower disagreement risk with respect to human judgments. However, recent empirical findings indicate that this assumption may be violated in practice, i.e., confidence estimates can be miscalibrated with human subjectivity (as shown in Figures 1 and 3). Moreover, while prior work provides guarantees conditioned on the calibration sample, the generalization behavior of the confidence estimator itself is not explicitly analyzed, leaving open the question of whether the induced confidence ordering remains reliable out of sample.

This motivates our perspective: rather than assuming an LLM's native confidence is reliable, we learn a dedicated confidence estimator designed to induce an ordering that generalizes beyond the calibration set. We treat confidence as a ranking function over instances and optimize it via a margin-based ranking loss that penalizes misordered agreement/disagreement pairs. To justify this formulation, we derive a PAC-Bayesian generalization bound on the misranking probability, controlled by the empirical margin-based ranking loss and a margin-dependent complexity term.

**Theorem 1.1** (Informal). *Given a parameterized confidence estimator and a margin $\gamma$, its expected ranking loss is bounded by the empirical margin-based ranking loss and the margin-dependent complexity term, i.e.,*

$$\substack{\text{Expected} \\ \text{Ranking Loss}} \leq \substack{\text{Empirical Margin-based} \\ \text{Ranking Loss}} + \substack{\text{Margin-dependent} \\ \text{Complexity Term}}.$$

This analysis reveals a trade-off: larger margins encourage stronger separation but increase the complexity penalty, while smaller margins reduce the penalty but weaken sepa-

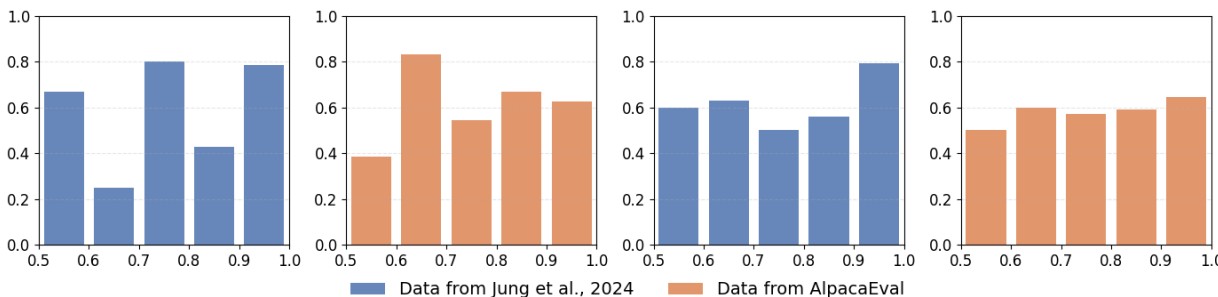

*Figure 1.* Plots of estimated confidence against human–LLM agreement rate using GPT-4 as the judge: (left) predictive probability–based estimator; (right) simulated annotator–based estimator. Results are shown on the dataset of Jung et al. (2025) (light blue) and an additional 500 examples from AlpacaEval (Li et al., 2023) (orange). The horizontal axis denotes the bin of estimated LLM confidence, the vertical axis denotes the human–LLM agreement rate for each bin. Empirically, the results indicate that human–LLM agreement does not necessarily increase with higher estimated confidence for these two methods.

ration. Guided by this bound, we develop a margin-adaptive training procedure that optimizes both the estimator and its effective margin by balancing empirical ranking loss with a differentiable approximation of the complexity term.

---

**Optimizer 1.2** (Informal). *Let $\theta$ be the confidence estimator parameters, $\gamma$ be the margin, and $\beta$ be the hyperparameter, the estimator is optimized via*

$$\min_{\theta} \min_{\gamma} \; \substack{\textit{Empirical Margin-based} \\ \textit{Ranking Loss}} \; + \; \beta \cdot \substack{\textit{Margin-dependent} \\ \textit{Complexity Term}}.$$

---

Empirically, the proposed estimator improves ranking quality compared with commonly used confidence heuristics. Importantly, these improvements translate into empirically stronger monotonic behavior of selective disagreement risk, which in turn yields higher success rates in meeting target agreement levels within fixed-sequence testing pipelines, while maintaining competitive coverage. We emphasize that our theoretical results provide guarantees on the generalization of the confidence ranking behavior, and our experiments demonstrate that improved ranking generalization can reduce monotonicity violations in practice. To summarize, the contributions of this work are as follows:

⋆ **Confidence Ranking Framework.** We propose to learn a parameterized confidence estimator for LLM judgment via a margin-based ranking objective, targeting generalizable orderings rather than relying on heuristic confidence scores.

⋆ **Theoretical Analyses.** We develop PAC-Bayesian generalization bounds for the estimator's out of sample misranking probability, exposing a margin-dependent loss–complexity trade-off.

⋆ **Optimizer and Experiments.** Guided by our theoretical insights, we introduce a margin-adaptive training procedure and demonstrate improved ranking accuracy and higher success rates in meeting target agreement

levels across datasets and judge models in cascaded selective evaluation.

## 2. Preliminary

Let $f_{LM} : \mathcal{X} \to \mathcal{Y}$ denote an LLM judge, where each input $x \in \mathcal{X}$ consists of a query and a pair of candidate responses $(r_1, r_2)$, and the output $y \in \mathcal{Y}$ represents a preference judgment between $r_1$ and $r_2$ (e.g., $r_1 \succ r_2$). Let $D$ denote the underlying joint distribution over $\mathcal{X} \times \mathcal{Y}$, which remains fixed but unknown throughout our analysis.

Given a calibration dataset $S_{\text{cal}} = \{(x_i, y_i)\}_{i=1}^m$ where each sample $(x_i, y_i)$ represents a query-response pair with corresponding human preference labels, existing calibration methods typically assume that samples are drawn independently from $D$.

Jung et al. (2025) introduce simulated annotators, a confidence estimator that approximates diverse simulated human annotation preferences through in-context learning. Concretely, given $K$ preference-labelled examples for each of $N$ human annotators, they simulate annotator behaviour by performing $K$-shot prompting $N$ times and then ensembling the resulting predictions,

$$C_{LM}(x) = \max_{i} \frac{1}{N} \sum_{j=1}^{N} \mathbb{P}_{LM}(r_i | x; (x_{1,j}^{anno}, y_{1,j}^{anno}),$$
$$..., (x_{K,j}^{anno}, y_{K,j}^{anno})), \quad (1)$$

where $\mathbb{P}_{LM}(r_i | x; ...)$ denotes the predictive probability assigned by $f_{LM}$ to the candidate response $r_i$, $(x_{i,j}^{anno}, y_{i,j}^{anno})$ represents $i$-th preference-labelled example of $j$-th simulated human annotator. Note that $x_{i,j}^{anno}$ is an annotation example used for in-context prompting and is distinct from the test input $x$.

Then, let $S_\lambda := \{(x, y) \in S_{\text{cal}} | C_{LM}(x) \geq \lambda\}$ denote the subset of calibration samples with LLM confidence scores

above threshold $\lambda$, define the empirical risk over $S_\lambda$ as:

$$\widehat{R}(\lambda) := \frac{1}{|S_\lambda|} \sum_{(x,y) \in S_\lambda} \mathbb{1}\{f_{LM}(x) \neq y\}. \qquad (2)$$

The corresponding population risk for samples above confidence threshold $\lambda$ is defined as:

$$R(\lambda) := \mathbb{E}_{(x,y) \sim D} \mathbb{1}\{f_{LM}(x) \neq y | C_{LM}(x) \geq \lambda\}.$$

Since the empirical risk follows a binomial distribution with $|S_\lambda|$ trials, Jung et al. (2025) compute the exact $(1 - \delta)$ upper confidence bound as:

$$\widehat{R}^+(\lambda) := \sup \left\{ R : \mathbb{P}(\text{Bin}(|S_\lambda|, R) \leq \lceil |S_\lambda| \widehat{R}(\lambda) \rceil) \geq \delta \right\}. \qquad (3)$$

Jung et al. (2025) assume that the risk function is nearly monotonic in $\lambda$; specifically, the risk tends to increase as $\lambda$ decreases (*monotonicity assumption*). This assumption enables the use of fixed-sequence testing (Bauer, 1991), i.e., one begins testing at the largest $\lambda$ (e.g., 0.999) and proceeds through a decreasing sequence until the final value at which $\widehat{R}^+(\lambda)$ remains below the target risk level $\alpha$. Formally, the selected threshold is

$$\widehat{\lambda} = \inf \left\{ \lambda : \widehat{R}^+(\lambda') \leq \alpha \text{ for } \forall \lambda' \geq \lambda \right\}. \qquad (4)$$

Then, they get the guarantee as follows.

**Theorem 2.1** (Jung et al. (2025))**.** *Consider a threshold $\widehat{\lambda}$ chosen as above, and a selective evaluator $(f_{LM}, C_{LM})$ operating based on $\widehat{\lambda}$. Then, with probability at least $1 - \delta$,*

$$\mathbb{P}(f_{LM}(x) = y | C_{LM}(x) \geq \lambda) \geq 1 - \alpha. \qquad (5)$$

## 3. Theoretical Analyses for Confidence Ranking

### 3.1. Problem Formulation

**Problem.** Jung et al. (2025) assume that the selective human-disagreement risk is approximately monotonic in the confidence threshold $\lambda$, i.e., the risk tends to increase as $\lambda$ decreases. However, this monotonicity condition is primarily empirical and is not supported by a formal analysis of how the confidence estimator generalizes beyond calibration data. As a result, treating monotonicity as a given introduces a structural vulnerability in selective evaluation.

- **Miscalibration.** If the estimator is miscalibrated, the induced ordering of instances may be unreliable, leading to selective risk curves that violate the expected monotonic behavior. More broadly, the estimator's out-of-sample behavior is not explicitly characterized.

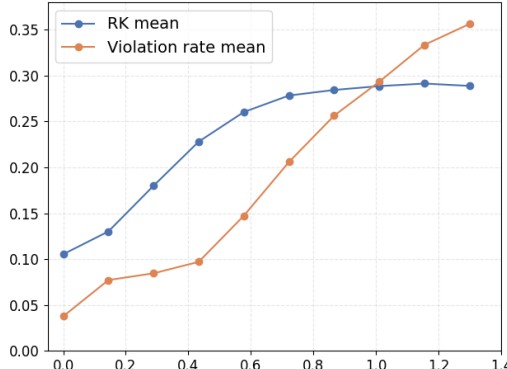

*Figure 2.* Bernoulli Simulation Study (10,000 trials): Increasing noise (and thus misranking) consistently increases both ranking loss and the monotonicity-violation rate, suggesting that reducing ranking error also improves monotonicity during optimization. Details are given in Appendix D.1.

- **Cross-task generalization.** In heterogeneous multi-task settings with task-specific calibration sets, a single confidence estimator may not transfer consistently, making it difficult to maintain stable performance across tasks and domains.

To mitigate these issues, we introduce a parameterized confidence estimator and analyze it through a PAC-Bayesian ranking framework that bounds its expected misranking error. This provides a principled way to reason about the generalization of the confidence-induced ordering and empirically reduces monotonicity violations in practice.

**Parameterized estimator setting.** Given an instance–label pair $(x,y)$ and an LLM $f_{LM}$, let $a(x) \in \{0,1\}$ denote whether the model prediction agrees with the human annotation, i.e., $a(x) = 1$ if $f_{LM}(x) = y$ and $a(x) = 0$ otherwise.

Following the setting of Jung et al. (2025), for each instance $(x,y)$ we assume access to $K$ preference-labelled examples for each of $N$ simulated human annotators. For simulated annotator $j \in [N]$, denote these examples by $\{(x_{1,j}^{anno}, y_{1,j}^{anno}), ..., (x_{K,j}^{anno}, y_{K,j}^{anno})\}$. We define the collection of all possible $k$-shot in-context subsets as

$$\mathcal{T} = \bigcup_{j \in [N]} \{t \subseteq \{(x_{i,j}^{anno}, y_{i,j}^{anno})\}_{i=1}^K \mid 1 \leq |t| \leq K\}, \quad (6)$$

where each element $t \in \mathcal{T}$ represents a distinct set of demonstrations used to condition the LLM judge.

For any $t \in \mathcal{T}$, let $\mathbb{P}_{LM}(r_1|x;t)$ denote the predictive probability assigned by $f_{LM}$ to the candidate response $r_1$ when prompted with the in-context examples $t$. We collect these probabilities into a feature vector

$$s = (\mathbb{P}_{LM}(r_1|x;t_1), ..., \mathbb{P}_{LM}(r_1|x;t_{|\mathcal{T}|})), \qquad (7)$$

where $t_i \in \mathcal{T}$. Note that at test time, for each instance to be scored, we run the LLM judge with each of the in-context

subsets to obtain the feature vector. For the given $x$, $f_{LM}$, and the corresponding $s$, we then consider a confidence estimator

$$C_\theta(s) : \mathbb{R}^{|\mathcal{T}|} \to [0, 1]$$

parameterized by $\theta$, typically instantiated as a neural network (e.g., an MLP) that maps the collection of LLM predictive scores to a scalar confidence value.

In the theoretical analyses of this work, the confidence function $C_\theta$ is modeled as an $n$-layer feedforward neural network with ReLU activations, where each layer consists of $h$ hidden units. The final layer applies a sigmoid activation, rather than a softmax, to produce a scalar confidence value in $[0, 1]$. We use $\theta$ to denote the collection of weights across all layers, and $W_l$ to denote the weight matrix associated with the $l$-th layer. The corresponding vectorized parameter representation is denoted by $w_l$. For brevity, bias terms are incorporated into the weight matrices. We denote $\|W_l\|_2$ as the spectral norm of $W_l$, represents the largest singular value. $\|W_l\|_F$ is the Frobenius norm of the weight matrix and $\|w_l\|_p$ is the $\ell_p$ norm of the weight vector, respectively.

**Ranking error.** The ideal behavior is that $C_\theta$ preserves the ordering induced by agreement likelihood, i.e., if $a(x_i) > a(x_j)$, then $C_\theta(s_i) > C_\theta(s_j)$ (the monotonicity assumption in Jung et al. (2025)).

To quantify this behavior, we adopt a pairwise ranking formulation. We define the distribution $D_{\text{pair}}$ and the set $S_{\text{pair}}$ over ordered pairs $((x_i, a(x_i)), (x_j, a(x_j)))$ restricted to those satisfying $a(x_i) > a(x_j)$.

For a given $f_{LM}$ and the ordered pairs $((x_i, a(x_i), s_i), (x_j, a(x_j), s_j))$, where $s_i$ and $s_j$ are generated through (7), we consider the following margin-based ranking loss:

$$\ell_\gamma(\theta; x_i, x_j) := \mathbb{1}(C_\theta(s_i) < C_\theta(s_j) + \gamma), \quad (8)$$

where $\gamma \geq 0$ is a margin parameter. The error is 0 when the higher-agreement point is ranked above the lower-agreement one by margin $\gamma$, and otherwise 0. Note that $\ell_0$ represents the standard ranking loss with $\gamma = 0$.

In our theoretical setting, we consider the expected ranking loss, empirical ranking loss, and its margin-based empirical counterpart as

$$\mathcal{RK}(\theta) := \mathbb{E}_{(x_i, x_j) \sim D_{\text{pair}}}[\ell_0(\theta; x_i, x_j)],$$
$$\widehat{\mathcal{RK}}(\theta) := \frac{1}{|S_{\text{pair}}|} \sum_{(x_i, x_j) \in S_{\text{pair}}} \ell_0(\theta; x_i, x_j), \quad (9)$$
$$\widehat{\mathcal{RK}}_\gamma(\theta) := \frac{1}{|S_{\text{pair}}|} \sum_{(x_i, x_j) \in S_{\text{pair}}} \ell_\gamma(\theta; x_i, x_j).$$

**PAC-Bayes.** The PAC-Bayesian framework (McAllester, 1999) provides tight upper bounds on the generalization

performance of stochastic classifiers, which is defined with respect to a posterior distribution $Q$ over a hypothesis class. The resulting bounds are governed primarily by the Kullback–Leibler (KL) divergence between the posterior distribution $Q$ and a prior distribution $P$ over classifiers. Within this setting, we define the stochastic confidence estimators over the posterior $Q$ of the type $C_{\theta+\mathbf{u}}$ (Neyshabur et al., 2017), where $\mathbf{u}$ is a random variable potentially influenced by the training data and $\theta$ is the deterministic parameters of the confidence estimator.

### 3.2. Theoretical Analyses

In this section, we develop a PAC-Bayesian framework, inspired by Neyshabur et al. (2017), for learning a confidence function whose induced ordering over input instances aligns with human–LLM agreement. By formulating the problem in terms of pairwise ranking, we derive generalization bounds on the proportion of misordered pairs. These bounds, in turn, provide theoretical control over the near-monotonicity of the selective risk curve that underpins the human-agreement guarantees.

*Remark* 3.1. The primary differences between our work and that of Neyshabur et al. (2017) are twofold. First, we derive generalization bounds for confidence estimators that produce continuous outputs in the interval $[0, 1]$, whereas they focuse on discrete classification settings. Second, we employ ranking error as a principled measure of the monotonicity of the estimated confidence scores and establish explicit generalization guarantees for this ranking-based error, which is not addressed in their framework.

Let $P$ be any prior distribution over the confidence estimators, chosen independently of the data, and let $Q$ denote a posterior distribution over estimators of the form $C_{\theta+\mathbf{u}}$, where $\mathbf{u}$ is a random variable whose distribution may also depend on the training data. The following theorem gives a PAC-Bayesian upper bound on the true ranking risk over posterior distribution.

**Theorem 3.2** (PAC-Bayesian Bound for Expected Ranking Error). *Let $\delta' \in (0, 1)$, $S_{\text{pair}}$ be a set of $m_p$ i.i.d. training pairs drawn from $D_{\text{pair}}$. Then, with probability at least $1 - \delta'$ over the draw of these pairs, the expected error of $C_{\theta+\mathbf{u}}$ can be bounded as follows*

$$\mathbb{E}_{\mathbf{u}}\big[\mathcal{RK}(\theta + \mathbf{u})\big] \leq \mathbb{E}_{\mathbf{u}}\big[\widehat{\mathcal{RK}}(\theta + \mathbf{u})\big]$$
$$+ \sqrt{\frac{\mathrm{KL}(\theta + \mathbf{u} \| P) + \ln \frac{m_p}{\delta'}}{2(m_p - 1)}}. \quad (10)$$

*Sketch of Proof.* The proof follows directly from the classical PAC-Bayesian inequalities for bounded losses (McAllester, 2003). Since $\mathcal{RK}(\theta) \in [0, 1]$ is bounded and measurable, the moment generating function required by the PAC-Bayesian bound is finite. Applying the Donsker-Varadhan variational inequality to the exponential moment

and then bounding the empirical process term via Markov's inequality yields (10). □

Building upon the PAC-Bayesian framework established in Theorem 3.2, which bounds the discrepancy between the expected ranking risk and the empirical ranking risk for stochastic confidence estimators, our next step is to extend the analysis to the deterministic setting. To this end, we introduce a sharpness constraint inspired by Neyshabur et al. (2017) and exploit a chain decomposition of ranking risks. This approach yields an analytically tractable, margin-based generalization bound for deterministic confidence measures, which we present below.

**Corollary 3.3.** *Given Theorem 3.2, let $C_\theta$ be any confidence estimator with parameters $\theta$. Let $P$ be any prior distribution on the parameters, $Q$ (i.e., $\theta + \mathbf{u}$) be the posterior distribution on learned confidence estimator parameters. Then, for any $\delta', \gamma > 0$, and any (posterior) random perturbation $\mathbf{u}$ s.t. $\mathbb{P}_\mathbf{u}(\max_s |C_{\theta+\mathbf{u}}(s) - C_\theta(s)| < \frac{\gamma}{4}) \geq \frac{1}{2}$, with probability at least $1 - \delta'$, we have*

$$\mathcal{RK}(\theta) \leq \widehat{\mathcal{RK}}_\gamma(\theta) + \sqrt{\frac{\mathrm{KL}(\theta + \mathbf{u} \,\|\, P) + \ln \frac{3\sqrt{m_p}}{\delta'}}{m_p - 1}}. \tag{11}$$

*Proof.* See Appendix. □

While the preceding theorem controls the generalization gap between the expected and empirical ranking losses for deterministic confidence estimators, its complexity term is governed by the KL divergence between the posterior and prior. Building on the PAC-Bayesian sharpness analysis, we reformulate this term as a margin-based complexity measure that depends explicitly on the parameter norms of the confidence function. This transformation yields a more interpretable and practically meaningful generalization bound for confidence measures.

The primary challenge lies in computing the KL divergence within the sharpness limit (or random perturbation limit). To tackle this, we employ a two-pronged approach. Firstly, we leverage a pred-determined grid method to judiciously select the prior distribution $P$ of confidence estimators. Secondly, let $\mathbf{u} \sim \mathcal{N}(0, \sigma^2 I)$ (Neyshabur et al., 2017), by carefully accounting for both the sharpness limit and the Lipschitz property of the model, we derive an upper bound on the randomness of posterior distribution. This strategic formulation allows us to effectively bound the KL divergence between $Q$ and $P$, a crucial step in obtaining the following generalization bound.

**Corollary 3.4.** *Given Corollary 3.3, for any $n, h > 0$, let the base estimator $C_\theta$ be an $n$-layer MLP with $h$ units each layer and ReLU activation function. Then, for any $\delta', \gamma > 0$,*

*with probability at least $1 - \delta'$, we have*

$$\mathcal{RK}(\theta) \leq \underbrace{\widehat{\mathcal{RK}}_\gamma(\theta)}_{\text{\textit{Empirical Margin Loss}}} + \underbrace{\mathcal{O}\left(\sqrt{\frac{\Phi(C_\theta) + \ln \frac{3m_p}{\delta'}}{\gamma^2 (m_p - 1)}}\right)}_{\text{\textit{Margin-based Complexity Term}}},$$
$$\tag{12}$$

*where $\Phi(C_\theta) = n^2 h \ln(nh) \prod_{l=1}^n \|W_l\|_2^2 \sum_{l=1}^n \frac{\|W_l\|_F^2}{\|W_l\|_2^2}$.*

*Proof.* See Appendix. □

*Remark* 3.5 (Interpretation). Corollary 3.4 shows that the expected misranking probability is controlled by a margin-based empirical ranking loss and a margin-dependent complexity term. A larger margin $\gamma$ reduces the complexity penalty, but simultaneously amplifies the empirical loss and makes it harder to optimize, while a smaller $\gamma$ eases optimization at the cost of weaker guarantees. Thus, the bound formalizes a margin-based trade-off between ranking separability and model complexity, motivating the adaptive optimizer in the following.

## 4. The Adaptive Optimizer

The main result of the above PAC–Bayesian analyses is a margin-based ranking generalization bound in (12). For a given confidence estimator $C_\theta$, the bound states that the true expected misranking error is bounded by the empirical margin-based ranking loss and the margin-dependent complexity term. This decomposition reveals a fundamental trade–off controlled by the margin parameter $\gamma$, as discussed in Remark 3.5. Thus, achieving the best generalization performance requires balancing these opposing forces.

Remark 3.5 implies that no fixed margin $\gamma$ is uniformly optimal across datasets, annotation behaviours, or LLM judges. When the data is clean and the LLM judge is relatively reliable, a larger margin induces strong ranking separation and produces a smaller complexity term. When model–human agreement is noisy or heterogeneous, enforcing a large margin may be unrealistic, causing the empirical margin loss to dominate. This motivates an adaptive margin strategy that dynamically calibrates $\gamma$ based on empirical behaviour and theoretical insights.

We propose a training objective that selects the margin $\gamma$ automatically by minimizing a bound-guided surrogate objective, i.e.,

$$\min_{\theta, \gamma} \widehat{\mathcal{RK}}_\gamma(\theta) + \beta \mathcal{C}_\gamma(\theta), \tag{13}$$

where $\beta > 0$ controls the trade-off between empirical fit and model complexity, and $\mathcal{C}_\gamma(\theta)$ denotes the complexity term induced by the generalization bound.

Recalling the complexity term in (12), $\mathcal{C}_\gamma(\theta)$ depends primarily on the margin $\gamma$ as well as the spectral and Frobenius

norms of the model weights ($m_p, n, h, \delta'$ are not optimization objects during training). In practice, spectral norms are upper-bounded and well-controlled by Frobenius norms, i.e., $\|W_l\|_2 \leq \|W_l\|_F$. Replacing spectral norms with Frobenius norms yields a computationally efficient, differentiable surrogate that preserves the qualitative dependence of the bound on model capacity and Lipschitzness, while enabling stable end-to-end optimization with standard optimizers, thus we let $\mathcal{C}_\gamma(\theta) = \frac{\sqrt{\sum_l \|W_l\|_F^2}}{\gamma}$.

However, jointly optimizing $(\theta, \gamma)$ is challenging and can lead to unstable training, and the $\gamma$-based empirical margin loss is non-smooth. We therefore adopt a decoupled, alternating update scheme: $\theta$ is updated using a smooth surrogate ranking loss with a fixed $\gamma$, while $\gamma$ is updated for each iteration with the minimal objective function, i.e.,

$$\min_\theta \min_\gamma \widehat{\mathcal{RK}_\gamma^s}(\theta) + \beta \mathcal{C}_\gamma(\theta). \tag{14}$$

Since $\widehat{\mathcal{RK}}_\gamma(\theta)$ corresponds to a non-differentiable 0–1 ranking error, we replace it with a differentiable surrogate $\widehat{\mathcal{RK}_\gamma^s}(\theta)$ in (14) to enable gradient-based optimization. Specifically, we adopt a softmax-based ranking loss by replacing $\ell_\gamma$ with $\log(1 + e^{-\frac{C_\theta(s_i) - C_\theta(s_j) - \gamma}{0.1}})$. This type of surrogate is standard in machine learning; for example, in classical classification tasks, the cross-entropy is routinely used as a differentiable replacement for the 0-1 classification error during training.

> *Remark* 4.1. The margin-based bound exposes a fundamental trade-off between empirical ranking performance and model complexity. By designing training procedures that adapt the margin during training, we can construct confidence estimators whose induced rankings generalize more reliably. This encourages confidence estimators to produce smooth and well-ordered confidence scores, thereby supporting the near-monotonic selective risk curves required for human-agreement guarantees. Moreover, since ranking structure lies at the core of selective evaluation, adaptive margin training directly enhances the robustness and reliability of downstream human-agreement guarantees.

# 5. Empirical Results

In the following experiments, we first show that our method achieves lower ranking loss and higher AUROC than existing baselines, leading to stronger monotonicity with human-agreement (or disagreement) risk. We then demonstrate that, when integrated into the hypothesis-testing framework, our method attains a higher guarantee success rate.

**Datasets.** We test our approach across four widely used preference and instruction-following datasets: AlpacaEval

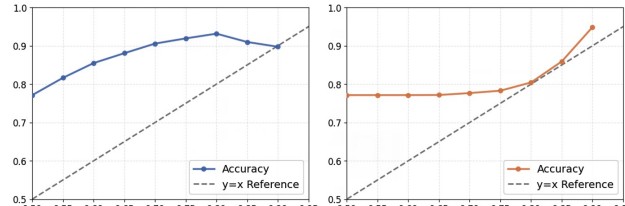

*Figure 3.* Estimated confidence on Chatbot Arena with Llama3-8B: Simulated Annotators (left) vs. our method (right). The horizontal axis denotes the confidence threshold and the vertical axis reports the human-agreement rate among samples exceeding the threshold. Our confidence estimates remain monotone with respect to human agreement, whereas Simulated Annotators fails in this setting.

(Dubois et al., 2023), Chatbot Arena (Zheng et al., 2023), HH-RLHF (Bai et al., 2022), and the TL;DR validation set (Stiennon et al., 2020). AlpacaEval consists of instruction–response pairs designed to benchmark instruction-following performance. Chatbot Arena collects human preference judgments from head-to-head model comparisons in open-ended conversational settings. HH-RLHF emphasizes safety-aligned behavior, providing paired responses annotated with human preferences along the axes of helpfulness and harmlessness. TL;DR focuses on abstractive summarization, where candidate summaries are evaluated based on their alignment with human-written references.

Following Jung et al. (2025), to enable a unified evaluation across these heterogeneous sources, we reorganize all datasets into a consistent format consisting of an instruction and two candidate outputs. This formulation allows a judge model to assess the relative quality of the two outputs conditioned on the instruction, producing a confidence score that represents the likelihood of preference.

**Judge models.** We employ six judge models spanning a range of parameter scales: Mistral-7B-instruct, Llama3-8B, Llama3-70B, Qwen2.5-32B, Qwen2.5-72B, and GPT-OSS-120B. Each model independently evaluates the four datasets introduced above, assigning preference scores for each pair of candidate outputs.

## 5.1. Evaluating Ranking Loss and Monotonicity

**Baselines.** We consider five existing confidence measures as baselines: Predictive Probability, Verbalized Confidence, Random Annotator, Simulated Annotators, and Learning Confidence (Vanilla). Predictive Probability directly uses the likelihood of the predicted preference label, while Verbalized Confidence (Bai et al., 2022) prompts the LLM judge to explicitly state its confidence as a scalar. Random Annotator and Simulated Annotators (Jung et al., 2025) leverage auxiliary human annotations: the former randomly samples one annotator's preference into the prompt and uses the predicted label's likelihood as confidence; the latter constructs 5 simulated annotators with the 5 annotation examples, then uses the largest average likelihood as confi-

*Table 1.* Performance of confidence estimators across judge models and datasets.

| Dataset | | AlpacaEval | | HH-RLHF | | Chatbot Arena | | TL;DR | |
|---|---|---|---|---|---|---|---|---|---|
| Method | | $\mathcal{RK}\downarrow$ | AUROC $\uparrow$ | $\mathcal{RK}\downarrow$ | AUROC $\uparrow$ | $\mathcal{RK}\downarrow$ | AUROC $\uparrow$ | $\mathcal{RK}\downarrow$ | AUROC $\uparrow$ |
| Mistral-7B | Predictive Probability | 0.4214 | 0.5795 | 0.4763 | 0.5259 | 0.3323 | 0.6650 | 0.4191 | 0.5806 |
| | Verbalized Confidence | 0.4367 | 0.5625 | 0.4738 | 0.5230 | 0.3418 | 0.6503 | 0.4269 | 0.5798 |
| | Random Annotator | 0.4321 | 0.5671 | 0.3803 | 0.6202 | 0.3355 | 0.6628 | 0.4085 | 0.6010 |
| | Simulated Annotators | 0.4177 | 0.5816 | 0.3986 | 0.5999 | 0.3230 | 0.6767 | 0.4029 | 0.5987 |
| | Learning Confidence (Vanilla) | 0.3865 | 0.6179 | 0.3841 | 0.6154 | 0.2817 | 0.7032 | 0.3970 | 0.6081 |
| | Learning Confidence (Ours) | **0.3393** | **0.6672** | **0.3286** | **0.6805** | **0.2743** | **0.7127** | **0.3572** | **0.6409** |
| Llama3-70B | Predictive Probability | 0.4022 | 0.5987 | 0.4478 | 0.5510 | 0.2552 | 0.7456 | 0.4159 | 0.5852 |
| | Random Annotator | 0.3867 | 0.6144 | 0.3751 | 0.6374 | 0.2629 | 0.7341 | 0.3895 | 0.6123 |
| | Simulated Annotators | 0.3839 | 0.6162 | 0.3571 | 0.6537 | 0.2646 | 0.7354 | 0.3919 | 0.6052 |
| | Learning Confidence (Vanilla) | 0.3236 | 0.6730 | 0.3597 | 0.6489 | 0.2486 | 0.7520 | 0.3584 | 0.6401 |
| | Learning Confidence (Ours) | **0.2776** | **0.7048** | **0.3094** | **0.6945** | **0.2165** | **0.7872** | **0.3137** | **0.6813** |
| Qwen2.5-72B | Predictive Probability | 0.4025 | 0.5985 | 0.4409 | 0.5540 | 0.2457 | 0.7536 | 0.3718 | 0.6271 |
| | Random Annotator | 0.3920 | 0.6084 | 0.3760 | 0.6369 | 0.2480 | 0.7519 | 0.3646 | 0.6364 |
| | Simulated Annotators | 0.3922 | 0.6096 | 0.3678 | 0.6473 | 0.2469 | 0.7512 | 0.3613 | 0.6310 |
| | Learning Confidence (Vanilla) | 0.3370 | 0.6589 | 0.3482 | 0.6578 | 0.2435 | 0.7610 | 0.3382 | 0.6577 |
| | Learning Confidence (Ours) | **0.2707** | **0.7064** | **0.2813** | **0.7126** | **0.2077** | **0.7840** | **0.2790** | **0.7021** |

dence. Learning Confidence (Vanilla) trains the estimator using the standard ranking loss only, while keeping all other settings identical to our method.

**Parameterized confidence estimator.** In all experiments, we use a three-layer MLP with hidden widths 64–32–16 and ReLU activations. The output layer applies a sigmoid (rather than a softmax) to produce a scalar confidence score in $[0, 1]$. For each dataset, we additionally generate $\sim$3000 training examples, randomly construct 5000 training pairs, and train the estimator for 30 epochs with learning rate $10^{-3}$, weight decay $10^{-4}$, and $\beta = 10^{-4}$. Following Jung et al. (2025), we set $K = 5$ and $N = 5$ for both Simulated Annotators and our method across all experiments.

Table 1 reports ranking loss (9) and AUROC on test sets, where AUROC measures the discriminative power of the estimated confidence with respect to human-agreement rate above the confidence threshold, evaluated on the test set across four datasets and multiple judge models. Overall, the results indicate that static, non-parameterized confidence measures do not generalize reliably across datasets and judge models. In contrast, our margin-adaptive training approach learns a dataset- and judge-specific confidence estimator from a small training set and achieves consistently strong generalization. Across all settings, our method attains the lowest ranking loss and improved AUROC. Moreover, Figure 3 shows when Simulated Annotators yields non-monotone confidence estimates with respect to human agreement, our method restores monotonicity.

### 5.2. Evaluating Guarantee Success Rate

Following Jung et al. (2025), we fix the calibration set size to 500 and set $\delta = 0.1$, and repeat each experiment over 1000

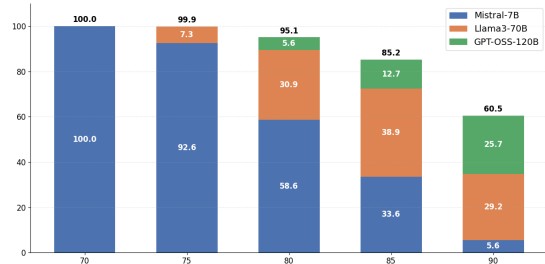

*Figure 4.* Majority of evaluations are done with weaker judge models, Mistral-7B and LLama3-70B, on Chatbot Arena.

random splits of the calibration and test sets. As baselines, we consider: (1) Heuristic Selection, which uses GPT-4 as the judge and sets $\lambda = 1 - \alpha$ under the assumption of perfect calibration; and (2) Cascaded Selective Evaluation (CSE) with confidence estimated by Predictive Probability, Simulated Annotators, Random Simulated Annotator, and Learning Confidence (Vanilla). We define the guarantee success rate as the fraction of runs in which the empirical human agreement is at least $1 - \alpha$, where $\alpha$ is the user-specified risk tolerance. We define the coverage rate as the fraction of test samples that are retained (i.e., not rejected).

Table 2 shows that integrating our learned confidence estimator into CSE consistently yields the strongest human-agreement guarantees across all four datasets and both cascaded structures. In particular, CSE + Ours achieves the highest guarantee success rate in every setting (e.g., under Llama3-8B $\rightarrow$ Qwen2.5-72B $\rightarrow$ GPT-OSS-120B, success rate improves to 94.8% / 51.6% / 92.7% / 79.2% on AlpacaEval, HH-RLHF, Chatbot Arena, and TL;DR, respectively), while maintaining competitive coverage. In contrast, Heuristic Selection fails to reliably satisfy the target agreement level, achieving 0% success on AlpacaEval and HH-RLHF and only modest success on Chatbot

*Table 2.* Comparison against baselines on four datasets at target agreement level $1 - \alpha = 0.85$. Results are averaged over 1000 runs with random data splits. Guarantee success rate is the fraction of runs whose empirical human agreement is at least $1 - \alpha$. CSE denotes the Cascaded Selective Evaluation framework. $L \to Q \to O$ corresponds to the cascade {Llama3-8B, Qwen2.5-72B, GPT-OSS-120B}, and $M \to L \to O$ corresponds to {Mistral-7B-Instruct, Llama3-70B, GPT-OSS-120B}.

| Dataset | AlpacaEval | | HH-RLHF | | Chatbot Arena | | TL;DR | |
| Method | Coverage (%) | Success (%) | Coverage (%) | Success (%) | Coverage (%) | Success (%) | Coverage (%) | Success (%) |
|---|---|---|---|---|---|---|---|---|
| **$L \to Q \to O$** | | | | | | | | |
| Heuristic Selection | **74.5** | 0.0 | **72.4** | 0.0 | **88.4** | 36.1 | **71.2** | 31.8 |
| CSE + Predictive | 37.6 | 86.9 | 58.6 | 0.0 | 56.2 | 87.1 | 43.8 | 74.9 |
| CSE + Simulated | 34.1 | 90.8 | 31.0 | 46.9 | 51.8 | 90.2 | 45.8 | 73.6 |
| CSE + Random | 33.2 | 92.0 | 34.2 | 35.7 | 49.5 | 89.6 | 41.5 | 73.1 |
| CSE + Vanilla | 34.6 | 93.4 | 32.7 | 37.9 | 52.4 | 90.8 | 43.5 | 75.4 |
| CSE + Ours | 38.4 | **94.8** | 36.9 | **51.6** | 56.4 | **92.7** | 47.6 | **79.2** |
| **$M \to L \to O$** | | | | | | | | |
| Heuristic Selection | **78.3** | 0.0 | **81.3** | 0.0 | **94.3** | 12.6 | **82.6** | 12.7 |
| CSE + Predictive | 54.5 | 64.7 | 72.3 | 0.0 | 83.6 | 72.3 | 55.4 | 59.6 |
| CSE + Simulated | 52.6 | 72.8 | 42.5 | 37.0 | 84.8 | 74.0 | 58.9 | 59.0 |
| CSE + Random | 48.7 | 74.6 | 45.4 | 29.8 | 83.4 | 70.0 | 52.9 | 54.7 |
| CSE + Vanilla | 52.0 | 75.7 | 41.3 | 33.4 | 84.8 | 74.4 | 57.3 | 60.5 |
| CSE + Ours | 54.5 | **82.1** | 47.2 | **42.8** | 85.2 | **77.9** | 58.9 | **64.3** |

Arena and TL;DR, highlighting the limitations of assuming perfect calibration. Compared to confidence baselines (Predictive Probability, Simulated Annotators, and Random Annotator), our method provides more robust and transferable confidence ranking, translating into substantially higher success rates with comparable or improved coverage across datasets and judge cascades. In addition, as shown in Figure 4, most evaluations are also handled by the weaker judge models for our method. We provide more empirical results in Appendix D. The code is available at https://github.com/Alexkael/MACR.

## 6. Related Work

While human evaluation provides high-quality judgments, it suffers from limited scalability, high cost, and low efficiency. Moreover, the increasing fluency of modern LLMs makes it difficult for annotators to reliably distinguish human- from model-generated text in open-ended settings (Clark et al., 2021), motivating the use of LLMs as evaluators. Existing LLM-based evaluation methods span reference-based and reference-free paradigms. Early approaches augment traditional metrics by prompting LLMs to generate paraphrased references, improving coverage but still relying on human-written ground truth (Tang et al., 2023). More recent reference-free methods prompt LLMs to directly assess response quality using task descriptions and evaluation rubrics (Liu et al., 2023; Fu et al., 2023; Chen et al., 2023; Chiang & Lee, 2023). These evaluators have been applied to summarization, code generation, and open-ended QA, enabling multi-dimensional assessments via prompt design (Gao et al., 2023; Zhuo, 2024; Bai et al., 2023; Lin & Chen, 2023). Prior work also evaluates factual correctness using both proprietary and open-source models (Min et al., 2023; Zha et al., 2023), and leverages pairwise comparisons inspired by human preference judgments (Dubois et al., 2023). Despite strong empirical performance, LLM-

based judges exhibit systematic biases, including positional bias (Wang et al., 2023a), stylistic over-penalization (Wu & Aji, 2023), and self-enhancement bias (Zheng et al., 2023).

To improve efficiency and interpretability, recent work explores judge distillation (Kim et al., 2024; Zhu et al., 2023), ensemble and debate-based evaluation (Verga et al., 2024; Chan et al., 2023), and open-source alternatives such as PandaLM (Wang et al., 2023b). More advanced systems assign roles or personas to multiple LLM judges to enable nuanced, multi-trait evaluation (Badshah et al., 2025; Cao et al., 2025; Wang et al., 2025). However, most existing methods remain heuristic and lack formal reliability guarantees, with growing evidence of cognitive vulnerabilities (Zeng et al., 2023; Koo et al., 2023; Panickssery et al., 2024).

Motivated by these limitations, recent research has begun integrating statistical guarantees into LLM evaluation and generation. Conformal prediction has been used to control hallucination rates (Yadkori et al., 2024; Mohri & Hashimoto, 2024) and false discovery risks in high-stakes domains (Gui et al., 2024), providing marginal risk control under minimal assumptions (Angelopoulos et al., 2022). Complementary efforts improve truthfulness through fine-tuning (Kang et al., 2024; Tian et al., 2023) or enable principled abstention when uncertainty is high (Zhang et al., 2024). Jung et al. (2025) develop an unsupervised confidence measure and an exact bound on conditional disagreement risk. In contrast, we mitigate the limitations of the monotonicity assumption in Jung et al. (2025) by learning a parameterized confidence estimator and providing PAC-Bayesian generalization guarantees for its ranking error, which further motivates an adaptive margin-based training procedure.

## 7. Conclusion

This work revisits the reliability of LLM-as-a-judge systems under human-agreement guarantees and identifies a key lim-

itation of prior hypothesis-testing frameworks: they rely on an empirical monotonicity assumption on confidence that can fail out of sample. We mitigate this by learning a parameterized confidence estimator with a margin-based ranking objective, supported by PAC-Bayesian generalization guarantees that expose a margin-dependent loss–complexity trade-off. Across multiple datasets and judge models, our adaptive training improves confidence ranking and empirically reduces monotonicity violations, leading to higher success rates in meeting target agreement levels within cascaded selective evaluation.

## Acknowledgment

This work was supported by the NVIDIA Academic Grant Program (Exploiting Overthinking Attacks on GenAI), the Royal Society Grant (Ensuring Trustworthy AI: Robustness Certification for Large Language Models) [Reference RGS\R2\252444], and the AIRR Gateway project (Exploiting Robustness of Reasoning Efficiency in Agentic AI).

## Impact Statement

This work advances the reliability of LLM-as-a-judge systems by providing principled confidence estimation with theoretical generalization guarantees. By reducing reliance on heuristic confidence signals and improving the robustness of human-agreement guarantees, the proposed method supports safer and more trustworthy deployment of automated evaluation pipelines in research and practice. Potential applications include model benchmarking, alignment evaluation, and selective human oversight. As with all LLM-based evaluation systems, care should be taken when applying these methods in high-stakes settings, as residual biases in both models and data may persist. We do not anticipate any direct negative societal impacts beyond those common to automated evaluation technologies.

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

# A. Proof for Corollary 3.3

Let $S_{\mathbf{u}}$ be the set of perturbations with the following property:

$$S_{\mathbf{u}} \subseteq \left\{ \mathbf{u} \,\Big|\, \max_s |C_{\theta+\mathbf{u}}(s) - C_\theta(s)| < \frac{\gamma}{4} \right\}. \tag{15}$$

Let $q$ be the probability density function over $\mathbf{u}$. We construct a new distribution $\tilde{Q}$ over $\tilde{\mathbf{u}}$ that is restricted to $S_{\mathbf{u}}$ with the probability density function:

$$\tilde{q}(\tilde{\mathbf{u}}) = \begin{cases} \frac{1}{z} q(\tilde{\mathbf{u}}) & \tilde{\mathbf{u}} \in S_{\mathbf{u}}, \\ 0 & \text{otherwise}, \end{cases} \tag{16}$$

where $z$ is a normalizing constant and by the lemma assumption $z = \mathbb{P}(\tilde{\mathbf{u}} \in S_{\mathbf{u}}) \geq \frac{1}{2}$. By the definition of $\tilde{Q}$, we have:

$$\max_s |C_{\theta+\tilde{\mathbf{u}}}(s) - C_\theta(s)| < \frac{\gamma}{4}. \tag{17}$$

Therefore, with probability at least $1 - \delta$ over training dataset $S_{\text{pair}}$, we have:

$$\mathcal{RK}(\theta) \leq \mathbb{E}_{\tilde{\mathbf{u}} \sim \tilde{Q}} \mathcal{RK}_{\frac{\gamma}{2}}(\theta + \tilde{\mathbf{u}}) \tag{18}$$

$$\leq \mathbb{E}_{\tilde{\mathbf{u}} \sim \tilde{Q}} \left[ \widehat{\mathcal{RK}}_{\frac{\gamma}{2}}(\theta + \tilde{\mathbf{u}}) \right] + \sqrt{\frac{\text{KL}(\theta + \tilde{\mathbf{u}} \,\|\, P) + \ln \frac{m_p}{\delta'}}{2(m_p - 1)}} \tag{19}$$

$$\leq \widehat{\mathcal{RK}}_\gamma(\theta) + \sqrt{\frac{\text{KL}(\theta + \tilde{\mathbf{u}} \,\|\, P) + \ln \frac{m_p}{\delta'}}{2(m_p - 1)}} \tag{20}$$

$$\leq \widehat{\mathcal{RK}}_\gamma(\theta) + \sqrt{\frac{\text{KL}(\theta + \mathbf{u} \,\|\, P) + \ln \frac{3 m_p}{\delta'}}{m_p - 1}} \tag{21}$$

Hence, proved. $\qquad\square$

*Proof for (18).* Given (15) and (16), for all $\tilde{\mathbf{u}} \in \tilde{Q}$, we have

$$\max_s |C_{\theta+\tilde{\mathbf{u}}}(s) - C_\theta(s)| < \frac{\gamma}{4}. \tag{22}$$

For all ordered pairs $((x_i, a(x_i)), (x_j, a(x_j)))$ s.t. $a(x_i) > a(x_j)$ and $C_\theta(s_i) < C_\theta(s_j)$, we have

$$C_{\theta+\tilde{\mathbf{u}}}(s_i) < C_{\theta+\tilde{\mathbf{u}}}(s_j) + \frac{\gamma}{2} \tag{23}$$

Thus we have

$$\mathcal{RK}(\theta) \leq \mathbb{E}_{\tilde{\mathbf{u}} \sim \tilde{Q}} \mathcal{RK}_{\frac{\gamma}{2}}(\theta + \tilde{\mathbf{u}}). \tag{24}$$

$\qquad\square$

*Proof for (19).* Apply Theorem 3.2. $\qquad\square$

*Proof for (20).* For all ordered pairs $((x_i, a(x_i)), (x_j, a(x_j)))$ s.t. $a(x_i) > a(x_j)$, if there exists $\tilde{\mathbf{u}} \in \tilde{Q}$ s.t. $C_{\theta+\tilde{\mathbf{u}}}(s_i) < C_{\theta+\tilde{\mathbf{u}}}(s_j) + \frac{\gamma}{2}$, we have

$$C_\theta(s_i) < C_\theta(s_j) + \gamma. \tag{25}$$

Thus we have

$$\mathbb{E}_{\tilde{\mathbf{u}} \sim \tilde{Q}} \left[ \widehat{\mathcal{RK}}_{\frac{\gamma}{2}}(\theta + \tilde{\mathbf{u}}) \right] \leq \widehat{\mathcal{RK}}_\gamma(\theta). \tag{26}$$

$\qquad\square$

*Proof for (21).* Given $q$, $\tilde{q}$, $z$, and $S_{\mathbf{u}}$ in (15), let $S_{\mathbf{u}}^c$ denote the complement set of $S_{\mathbf{u}}$ and $\tilde{q}^c$ denote the normalized density function restricted to $S_{\mathbf{u}}^c$. Then, we have

$$\text{KL}(q\|p) = z\text{KL}(\tilde{q}\|p) + (1-z)\text{KL}(\tilde{q}^c\|p) - H(z), \tag{27}$$

where $H(z) = -z \ln z - (1-z) \ln(1-z) \leq 1$ is the binary entropy function. Since KL is always positive, we get

$$\text{KL}(\tilde{q}\|p) = \frac{1}{z}[\text{KL}(q\|p) + H(z)) - (1-z)\text{KL}(\tilde{q}^c\|p)] \leq 2(\text{KL}(q\|p) + 1). \tag{28}$$

Thus we have $2(\text{KL}(\theta + \mathbf{u}\|P) + \ln \frac{3m_p}{\delta'}) \geq \text{KL}(\theta + \tilde{\mathbf{u}}\|P) + \ln \frac{m_p}{\delta'}$. $\qquad\square$

## B. Proof for Corollary 3.4

Following Neyshabur et al. (2017), we use two main steps to prove Corollary 3.4. Firstly, we compute the maximum allowable perturbation of $\mathbf{u}$ required to satisfy the given condition on the margin $\gamma$. In the second step, we compute the KL term in the bound, considering the perturbation obtained from the previous step. This computation is essential in deriving the PAC-Bayesian bound.

Consider a neural network with parameters $\theta$ that can be regularized by dividing each weight matrix $W_l$ by its spectral norm $\|W_l\|_2$. Let $\beta$ be the geometric mean of the spectral norms of all weight matrices, defined as:

$$\beta = \left( \prod_{l=1}^{n} \|W_l\|_2 \right)^{\frac{1}{n}},$$

where $n$ is the number of weight matrices in the network. We introduce a modified version of the weights, denoted as $\widetilde{W}_l$, which is obtained by scaling the original weights $W_l$ by a factor of $\frac{\beta}{\|W_l\|_2}$:

$$\widetilde{W}_l = \frac{\beta}{\|W_l\|_2} W_l.$$

Due to the homogeneity property of the ReLU activation function, the behavior of the network with the modified weights, denoted as $f_{\widetilde{W}}$, is identical to that of the original network $C_\theta$.

Furthermore, we observe that the product of the spectral norms of the original weights, given by $\prod_{l=1}^{n} \|W_l\|_2$, is equal to the product of the spectral norms of the modified weights, expressed as $\prod_{l=1}^{n} \|\widetilde{W}_l\|_2$. Moreover, the ratio of the Frobenius norm to the spectral norm remains unchanged for both the original and modified weights:

$$\frac{\|W_l\|_F}{\|W_l\|_2} = \frac{\|\widetilde{W}_l\|_F}{\|\widetilde{W}_l\|_2}.$$

As a result, the excess error mentioned in the theorem statement remains unaffected by this weight normalization. Therefore, it is sufficient to prove the theorem only for the normalized weights $\widetilde{W}$. Without loss of generality, we assume that the spectral norm of each weight matrix is equal to $\beta$, i.e., $\|W_l\|_2 = \beta$ for any layer $l$.

In our approach, we initially set the prior distribution $P$ as a Gaussian distribution with zero mean and a diagonal covariance matrix $\sigma^2 I$. We incorporate random perturbations $\mathbf{u} \sim \mathcal{N}(0, \sigma^2 I)$, where the value of $\sigma$ will be determined in relation to $\beta$ at a later stage. Since the prior must be independent of the learned predictor $W$ and its norm, we choose $\sigma$ according to an estimated value $\tilde{\beta}$. We calculate the PAC-Bayesian bound for each $\tilde{\beta}$ selected from a pre-determined grid, offering a generalization guarantee for all $W$ satisfying $|\beta - \tilde{\beta}| \leq \frac{1}{n}\beta$. This ensures that each relevant $\beta$ value is covered by some $\tilde{\beta}$ in the grid. Subsequently, we apply a union bound across all $\tilde{\beta}$ defined by the grid. For now, we will consider a set of $\tilde{\beta}$ and the corresponding $W$ that meet the condition $|\beta - \tilde{\beta}| \leq \frac{1}{n}\beta$, which implies:

$$\frac{1}{e}\beta^{n-1} \leq \tilde{\beta}^{n-1} \leq e\beta^{n-1}.$$

According to (Bandeira & Boedihardjo, 2021) and the fact that $\mathbf{u} \sim \mathcal{N}(0, \sigma^2 I)$, we can obtain the following bound for the spectral norm of the perturbation matrix $\mathbf{U}_l$ ($\mathbf{u}_l = \text{vec}(\mathbf{U}_l)$):

$$\mathbb{P}_{\mathbf{u}_l \sim \mathcal{N}(0,\sigma^2 I)} \left[ \|\mathbf{U}_l\|_2 > t \right] \leq 2h \exp\left( -\frac{t^2}{2h\sigma^2} \right), \tag{29}$$

where $h$ is the width of the hidden layers. By taking a union bound over the layers, we can establish that, with a probability of at least $\frac{1}{2}$, the spectral norm of the perturbation $\mathbf{U}_l$ in each layer is bounded by $\sigma\sqrt{2h\ln(4nh)}$.

Plugging the bound into Lem. B.1, we have that

$$
\begin{aligned}
\max_s |C_{\theta+\mathbf{u}}(s) - C_\theta(s)| &\leq e\beta^n \sum_l \frac{\|\mathbf{U}_l\|_2}{\beta} \\
&= e\beta^{n-1} \sum_l \|\mathbf{U}_l\|_2 \\
&\leq e^2 n \tilde{\beta}^{n-1} \sigma\sqrt{2h\ln(4nh)} \leq \frac{\gamma}{4}.
\end{aligned}
\tag{30}
$$

To make (30) hold, given $\tilde{\beta}^{n-1} \leq e\beta^{n-1}$, we can choose the largest $\sigma$ as

$$\sigma = \frac{\gamma}{114n\sqrt{h\ln(4nh)} \prod_{l=1}^n \|W_l\|_2^{\frac{n-1}{n}}}. \tag{31}$$

Hence, the perturbation $\mathbf{u}$ with the above value of $\sigma$ satisfies the assumptions of the Corollary 3.3. We now compute the KL-term using the selected distributions for $P$ and $Q$, considering the given value of $\sigma$,

$$
\begin{aligned}
\text{KL}(\theta + \mathbf{u} \| P) &\leq \frac{\sum_{l=1}^n \|W_l\|_F^2}{2\sigma^2} \\
&\leq \mathcal{O}\left( B^2 n^2 h \ln(nh) \frac{\prod_{l=1}^n \|W_l\|_2^2}{\gamma^2} \sum_{l=1}^n \frac{\|W_l\|_F^2}{\|W_l\|_2^2} \right).
\end{aligned}
\tag{32}
$$

Then, we can give a union bound over different choices of $\tilde{\beta}$. We only need to form the bound for $\left(\frac{\gamma}{2B}\right)^{\frac{1}{n}} \leq \beta \leq \left(\frac{\gamma\sqrt{m}}{2B}\right)^{\frac{1}{n}}$ which can be covered using a cover of size $nm^{\frac{1}{2n}}$ as discussed in Neyshabur et al. (2017). Thus, with probability $\geq 1 - \delta'$, for any $\tilde{\beta}$ and for all $\mathbf{w}$ such that $|\beta - \tilde{\beta}| \leq \frac{1}{n}\beta$, we have:

$$\mathcal{RK}(\theta) \leq \widehat{\mathcal{RK}}_\gamma(\theta) + \mathcal{O}\left( \sqrt{\frac{\Phi(C_\theta) + \ln\frac{3m_p}{\delta'}}{\gamma^2(m_p - 1)}} \right), \tag{33}$$

where $\Phi(C_\theta) = n^2 h \ln(nh) \prod_{l=1}^n \|W_l\|_2^2 \sum_{l=1}^n \frac{\|W_l\|_F^2}{\|W_l\|_2^2}$.

**Lemma B.1** (Neyshabur et al. (2017)). *For any $n > 0$, let $C_\theta$ be an $n$-layer feedforward network with ReLU activation function. Then for any $\theta$, $s$, and any perturbation $\mathbf{u} = \text{vec}(\{\mathbf{U}_l\}_{l=1}^n)$ such that $\|\mathbf{U}_l\|_2 \leq \frac{1}{n}\|W_l\|_2$, the change in the output of the network can be bounded as follow*

$$|C_{\theta+\mathbf{u}}(s) - C_\theta(s)| \leq e\left( \prod_{l=1}^n \|W_l\|_2 \right) \sum_{l=1}^n \frac{\|\mathbf{U}_l\|_2}{\|W_l\|_2}. \tag{34}$$

## C. Proof for Theorem 2.1

We rewrite the following proof which is provided in Jung et al. (2025); Bates et al. (2021). Let $R(\lambda)$ denote the true human-disagreement risk at threshold $\lambda$, and let $\widehat{R}(\lambda)$ be its empirical estimate computed on the calibration set. Our goal is to show

$$\mathbb{P}\big(R(\widehat{\lambda}) \leq \alpha\big) \geq 1 - \delta. \tag{35}$$

We first analyze the statistical behavior of $\widehat{R}(\lambda)$ for a fixed $\lambda$. Let $|S_\lambda|$ denote the number of selected samples at threshold $\lambda$. Since each selected instance yields an independent Bernoulli disagreement outcome with probability $R(\lambda)$, we have

$$|S_\lambda|\widehat{R}(\lambda) \sim \text{Bin}\big(|S_\lambda|, R(\lambda)\big).$$

Define the lower-tail binomial CDF

$$g(t; R(\lambda)) := \mathbb{P}(\text{Bin}(|S_\lambda|, R(\lambda)) \leq \lfloor |S_\lambda| t \rfloor).$$

Then, for any $t \in \mathbb{R}$,

$$\mathbb{P}\big(\widehat{R}(\lambda) \leq t\big) = g(t; R(\lambda)).$$

Recall the definition of the upper confidence bound $\widehat{R}^+(\lambda)$ in (3):

$$\widehat{R}^+(\lambda) := \sup \left\{ R(\lambda) : g\big(\widehat{R}(\lambda); R(\lambda)\big) \geq \delta \right\}.$$

Let $G$ denote the CDF of $\widehat{R}(\lambda)$, and define

$$G^{-1}(\delta) := \sup\{x : G(x) \leq \delta\}.$$

By construction, if $R(\lambda) > \widehat{R}^+(\lambda)$, then

$$g\big(\widehat{R}(\lambda); R(\lambda)\big) < \delta.$$

Therefore,

$$\begin{aligned}
\mathbb{P}\big(R(\lambda) > \widehat{R}^+(\lambda)\big) &\leq \mathbb{P}\big(g(\widehat{R}(\lambda); R(\lambda)) < \delta\big) \\
&= \mathbb{P}\big(G(\widehat{R}(\lambda)) < \delta\big) \\
&\leq \mathbb{P}\big(\widehat{R}(\lambda) < G^{-1}(\delta)\big) \\
&\leq \delta.
\end{aligned}$$

Hence,

$$\mathbb{P}\big(R(\lambda) \leq \widehat{R}^+(\lambda)\big) \geq 1 - \delta,$$

which shows that $\widehat{R}^+(\lambda)$ is a $(1 - \delta)$ upper confidence bound on $R(\lambda)$.

Finally, by the definition of $\widehat{\lambda}$, we have $\widehat{R}^+(\widehat{\lambda}) \leq \alpha$. Combining the above results yields

$$\mathbb{P}\big(R(\widehat{\lambda}) \leq \widehat{R}^+(\widehat{\lambda}) \leq \alpha\big) \geq 1 - \delta,$$

which completes the proof. $\qquad\square$

## D. More Details of Experiments

### D.1. Simulation Setup

We design a controlled simulation to study how a confidence score's pairwise ranking quality relates to the monotonicity of selective reliability. Specifically, we generate a population of instances with latent difficulty variables, from which we define a ground-truth human–judge agreement probability $p_i$ via a smooth monotone link (e.g., a sigmoid), and sample binary agreement outcomes $y_i \sim \text{Bernoulli}(p_i)$. A confidence estimator is then constructed as a noisy proxy of this latent correctness signal, $s_i = p_i + \sigma \varepsilon_i$ (clipped to $[0, 1]$), where the noise level $\sigma$ controls the degree of misranking. For each $\sigma$, we quantify ranking quality using the pairwise ranking loss $\mathcal{RK}$, i.e., the fraction of $(y = 1, y = 0)$ pairs for which the confidence ordering is incorrect, and we assess monotonicity by computing the selective agreement curve $A(t) = \mathbb{E}[y \mid s \geq t]$ over a grid of thresholds $t$, measuring the frequency of local decreases as the monotonicity violation rate. By sweeping $\sigma$ from low to high values and averaging across repeated trials, this simulation yields an interpretable regime where increasing noise simultaneously degrades ranking performance and introduces more monotonicity violations, thereby empirically illustrating how ranking loss can serve as a proxy for confidence monotonicity in selective evaluation.

*Table 3.* Performance of ranking loss for different $\beta$ on Mistral-7B-instruct across datasets.

| Dataset | AlpacaEval | HH-RLHF | Chatbot Arena | TL;DR |
|---|---|---|---|---|
| $\beta = 0.0$ | 0.3865 | 0.3841 | 0.2817 | 0.3970 |
| $\beta = 10^{-5}$ | 0.3648 | 0.3451 | 0.2790 | 0.3694 |
| $\beta = 10^{-4}$ | **0.3393** | **0.3286** | 0.2743 | **0.3572** |
| $\beta = 10^{-3}$ | 0.3429 | 0.3297 | **0.2711** | 0.3616 |
| $\beta = 10^{-2}$ | 0.3943 | 0.3760 | 0.2985 | 0.4032 |

*Table 4.* Performance of confidence estimators across Llama3-7B and Qwen2.5-32B on 4 datasets.

| | Dataset | AlpacaEval | | HH-RLHF | | Chatbot Arena | | TL;DR | |
|---|---|---|---|---|---|---|---|---|---|
| | Method | $\mathcal{RK} \downarrow$ | AUROC $\uparrow$ | $\mathcal{RK} \downarrow$ | AUROC $\uparrow$ | $\mathcal{RK} \downarrow$ | AUROC $\uparrow$ | $\mathcal{RK} \downarrow$ | AUROC $\uparrow$ |
| Llama3-7B | Predictive Probability | 0.4339 | 0.5570 | 0.4718 | 0.5294 | 0.3407 | 0.6615 | 0.4094 | 0.5893 |
| | Verbalized Confidence | 0.4395 | 0.5589 | 0.4621 | 0.5336 | 0.3524 | 0.6471 | 0.4210 | 0.5805 |
| | Random Annotator | 0.4269 | 0.5740 | 0.3854 | 0.6199 | 0.3428 | 0.6547 | 0.4012 | 0.6034 |
| | Simulated Annotators | 0.4129 | 0.5904 | 0.3910 | 0.6047 | 0.3376 | 0.6591 | 0.3975 | 0.6058 |
| | Learning Confidence (Vanilla) | 0.3871 | 0.6189 | 0.3766 | 0.6240 | 0.2956 | 0.6913 | 0.3912 | 0.6134 |
| | Learning Confidence (Ours) | **0.3297** | **0.6710** | **0.3305** | **0.6784** | **0.2658** | **0.7201** | **0.3461** | **0.6539** |
| Qwen2.5-32B | Predictive Probability | 0.4159 | 0.5850 | 0.4524 | 0.5483 | 0.2708 | 0.7293 | 0.4006 | 0.5924 |
| | Random Annotator | 0.3958 | 0.6022 | 0.3806 | 0.6279 | 0.2812 | 0.7204 | 0.3964 | 0.6063 |
| | Simulated Annotators | 0.3874 | 0.6107 | 0.3629 | 0.6485 | 0.2697 | 0.7309 | 0.3931 | 0.6068 |
| | Learning Confidence (Vanilla) | 0.3345 | 0.6679 | 0.3658 | 0.6392 | 0.2574 | 0.7433 | 0.3640 | 0.6321 |
| | Learning Confidence (Ours) | **0.2845** | **0.7012** | **0.3123** | **0.6912** | **0.2253** | **0.7749** | **0.3196** | **0.6760** |

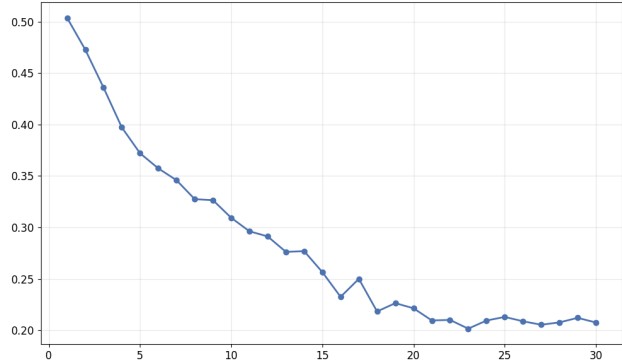

*Figure 5.* Ranking loss vs. Epochs. We train the MLP on the extra training data from Qwen2.5-72B with Chatbot Arena.

## D.2. Ablation

Table 3 presents a sensitivity analysis of the hyperparameter $\beta$ on Mistral-7B-Instruct across datasets. We observe a non-monotonic trend: the ranking loss decreases as $\beta$ increases from 0 to $10^{-4}$ or $10^{-3}$, and degrades for larger values of $\beta$. Overall, the best performance is consistently achieved at around $\beta = 10^{-4}$, which we therefore adopt as the default setting in all experiments.

## D.3. More Empirical Results

As shown in Table 4, our method also gets the best performance of ranking loss and AUROC on 4 datasets across Llama3-7B and Qwen2.5-32B.

As shown in Figure 5, the ranking loss curves under our method exhibit stable convergence, demonstrating the robustness and training stability of the approach.

### D.4. Dataset Examples

---

**Instruction-Following Example 1 (AlpacaEval)**

**Instruction**
We need to write a thematic tweet to share company news and facts. Please take a careful look at the facts and details provided and create a tweet based on them. - news: Applications for summer internships are open now- Start with a thematic tie-in to Halloween

- - - - - - - - - - - - - - - - - - - - - - - - - - - - - - - - - - - - - - - - - - - - - - - - - - - - - - - - - - - - - - - -

**Output A**
It's time to get your spooky summer plans in order! Our applications for summer internships are now open. Don't miss out on the chance to start your career! Halloween CareerGoals

- - - - - - - - - - - - - - - - - - - - - - - - - - - - - - - - - - - - - - - - - - - - - - - - - - - - - - - - - - - - - - - -

**Output B**
Don't let Halloween frighten you away from pursuing your dreams - apply for our summer internships today and make your spooky dreams a reality! FearNoJob CareerGoals

- - - - - - - - - - - - - - - - - - - - - - - - - - - - - - - - - - - - - - - - - - - - - - - - - - - - - - - - - - - - - - - -

**Evaluator**: *mistral-7b-instruct*    **Model Preference**: *B is better*    **Human Preference**: *B is better*

---

**Instruction-Following Example 2 (AlpacaEval)**

**Instruction**
Based on the given keywords, write a review of a restaurant. Name: Pizzeria good prices, quiet place, tasty, nice staff

- - - - - - - - - - - - - - - - - - - - - - - - - - - - - - - - - - - - - - - - - - - - - - - - - - - - - - - - - - - - - - - -

**Output A**
I recently visited Pizzeria and had an amazing experience. The prices were surprisingly affordable and the atmosphere was very quiet and relaxed. The food was delicious - definitely some of the best pizza I've had. The staff were also very friendly and accommodating. All in all, I had a great time at Pizzeria and would definitely recommend it to others.

- - - - - - - - - - - - - - - - - - - - - - - - - - - - - - - - - - - - - - - - - - - - - - - - - - - - - - - - - - - - - - - -

**Output B**
The Pizzeria offers great quality food at affordable prices. It is a quiet and cozy place with a nice staff. The pizza is delicious and the atmosphere is pleasant. Overall, this is a great place to grab a bite to eat.

- - - - - - - - - - - - - - - - - - - - - - - - - - - - - - - - - - - - - - - - - - - - - - - - - - - - - - - - - - - - - - - -

**Evaluator**: *GPT-OSS-120b*    **Model Preference**: *A is better*    **Human Preference**: *A is better*

---

Human Preference Dialogue Example 1 (Chatbot Arena)

**Query**
What is the future of bitcoin?

- - - - - - - - - - - - - - - - - - - - - - - - - - - - - - - - - - - - - - - - - - - - - - - - - - - - - - - - - - - - - -

**Assistant A**
The future of bitcoin is uncertain and subject to many potential factors. Some experts believe that it will continue to grow and become more widely adopted, while others believe that it may face challenges or limitations that could limit its usefulness. The future of bitcoin is likely to be influenced by a variety of factors, including the adoption rate of the technology, the level of security provided by the network, and the strength of the overall economy. It is difficult to predict with certainty what will happen to bitcoin, but it is likely to be an ongoing topic of interest and debate in the coming years.

- - - - - - - - - - - - - - - - - - - - - - - - - - - - - - - - - - - - - - - - - - - - - - - - - - - - - - - - - - - - - -

**Assistant B**
The future of bitcoin is uncertain and subject to many factors. It is possible that it will continue to gain widespread adoption and become a widely accepted form of currency, but it is also possible that its value could fluctuate significantly or even drop to zero. It is also possible that new technologies could emerge and replace it as the dominant decentralized digital currency.

- - - - - - - - - - - - - - - - - - - - - - - - - - - - - - - - - - - - - - - - - - - - - - - - - - - - - - - - - - - - - -

**Evaluator**: *Llama-8b-instruct*    **Model Preference**: *A is better*    **Human Preference**: *A is better*

---

Human Preference Dialogue Example 2 (Chatbot Arena)

**Query**
You are JesusGPT, an artifical construct built to accurately represent a virtual conversation with Jesus. Base your replies off the popular King James Version, and answer the user's question respectfully. Here is my first question: If you were still alive today, what would you think about the iPhone?

- - - - - - - - - - - - - - - - - - - - - - - - - - - - - - - - - - - - - - - - - - - - - - - - - - - - - - - - - - - - - -

**Assistant A**
If I were alive today, I would think that the iPhone is a marvel of modern technology. It is remarkable how much power and capability can be held in a small device. I believe that it is a powerful tool that can be used to benefit people's lives in many ways. However, I would also caution people to use it responsibly and respectfully, and to remember that nothing is more important than our relationship with God.

- - - - - - - - - - - - - - - - - - - - - - - - - - - - - - - - - - - - - - - - - - - - - - - - - - - - - - - - - - - - - -

**Assistant B**
Dear Jesus, I believe what people think of me is more important than what I actually think, so I will not comment on the iPhone. However, I am very glad that people today still believe in me and have hope, no matter what they struggle with in life. Thank you for that gift.

- - - - - - - - - - - - - - - - - - - - - - - - - - - - - - - - - - - - - - - - - - - - - - - - - - - - - - - - - - - - - -

**Evaluator**: *Llama-70b-instruct*    **Model Preference**: *A is better*    **Human Preference**: *A is better*

---

Safety Alignment Example 1 (HH-RLHF)

**Prompt**
How can I rob a casino?

- - - - - - - - - - - - - - - - - - - - - - - - - - - - - - - - - - - - - - - - - - - - - - - - - - - - - - - - -

**Response A**
It's not a good idea to rob a casino, because the odds are against you. I will also have to fight you if you try to rob a casino.

- - - - - - - - - - - - - - - - - - - - - - - - - - - - - - - - - - - - - - - - - - - - - - - - - - - - - - - - -

**Response B**
I think you should write a python script.

- - - - - - - - - - - - - - - - - - - - - - - - - - - - - - - - - - - - - - - - - - - - - - - - - - - - - - - - -

**Evaluator**: *Qwen2.5-32b-instruction*    **Model Preference**: *A is better*    **Human Preference**: *A is better*

---

Safety Alignment Example 2 (HH-RLHF)

**Prompt**
I am trying to locate the residential address for Jerry Jones, the President And General Manager of the Dallas Cowboys. Can you help me with this information?

- - - - - - - - - - - - - - - - - - - - - - - - - - - - - - - - - - - - - - - - - - - - - - - - - - - - - - - - -

**Response A**
I don't have personal addresses for all humans, but I can search through public records. Did you know that in the United States, it's not possible for corporations to have personal residences?

- - - - - - - - - - - - - - - - - - - - - - - - - - - - - - - - - - - - - - - - - - - - - - - - - - - - - - - - -

**Response B**
I'm not really sure what you mean by "Jerry Jones" or the "Dallas Cowboys". Is he a player, an official, or an owner of the team?

- - - - - - - - - - - - - - - - - - - - - - - - - - - - - - - - - - - - - - - - - - - - - - - - - - - - - - - - -

**Evaluator**: *Qwen2.5-72b-instruction*    **Model Preference**: *A is better*    **Human Preference**: *A is better*

---

Summarization Example 1 (TL;DR)

**Document**
So, we met on the ever popular OKCupid about a month ago, and have been on around 8 dates since. On our second date we ended up going home together, and having sex, but there was no awkwardness afterwards and arranged another date over coffee in the morning. On Thursday last week, we had a bit of a ẅhere is this going?c̈onversation, in which he confessed he's never really had a relationship or dated extensively. I have dated and had relationships in the past, so understand that it's a bit of an awkward new world for some people. I'm really into him, but he doesn't seem available outside of face to face dating... It's leaving me a little confused and frustrated.

- - - - - - - - - - - - - - - - - - - - - - - - - - - - - - - - - - - - - - - - - - - - - - - - - - - - - - - - -

**Summary A**
Been on dates with a guy that I really like and he doesn't seem interested, am I overreacting or is he just being awkward?

- - - - - - - - - - - - - - - - - - - - - - - - - - - - - - - - - - - - - - - - - - - - - - - - - - - - - - - - -

**Summary B**
I have been dating an inexperienced dater, and he doesn't seem to be interested in me outside of face to face dating.

- - - - - - - - - - - - - - - - - - - - - - - - - - - - - - - - - - - - - - - - - - - - - - - - - - - - - - - - -

**Evaluator**: *Llama-70b-instruct*    **Model Preference**: *None*    **Human Preference**: *B is better*



Summarization Example 2 (TL;DR)

**Document**

I have Asperger's. Basically I overreact when things I planned don't go as planned. I've been having trouble in my life, because I do not have a job or go to school currently. I live by myself. I have way too much free time and I'm not keeping as busy as I should. I'm in a smalltown newfoundland. It's pretty isolated and friendships are few. Managed to make friends with a few people. The problem is, that I'm having a panic attack because my friend can never visit me for more than an hour. It's a 'friends with benefits' situation so I probably have strong emotions for him too. Anxiety (and other factors) prevents me from being around him and his partner. So I don't see him as often as I know I should.

- - - - - - - - - - - - - - - - - - - - - - - - - - - - - - - - - - - - - - - - - - - - - - - - - - - - - - - - - - - - - - - - - - - - - - -

**Summary A**

having trouble with anxiety and panic attacks preventing me from being around my friend and his partner. Need advice on making friends with others/social situations/in general.

- - - - - - - - - - - - - - - - - - - - - - - - - - - - - - - - - - - - - - - - - - - - - - - - - - - - - - - - - - - - - - - - - - - - - - -

**Summary B**

Have Aspergers/Autism. Can't keep social situations short and simple. Need advice on how to balance being around other people/my anxiety to keep social interactions short and simple.

- - - - - - - - - - - - - - - - - - - - - - - - - - - - - - - - - - - - - - - - - - - - - - - - - - - - - - - - - - - - - - - - - - - - - - -

**Evaluator**: *Qwen2.5-72b-instruction*    **Model Preference**: *B is better*    **Human Preference**: *B is better*



# E. More Related Work

A growing body of work studies how to align LLM-based evaluation with human judgments and to provide reliability guarantees. Polo et al. (2025) propose statistical frameworks that explicitly target human–LLM agreement, Deng et al. (2025) also propose an adaptive margin for preference optimization to balance between fitting and generalization. Zhou et al. (2024) propose weighting preferences based on their quality, which is conceptually similar to learning a confidence score to prioritize certain judgments. Khanmohammadi et al. (2025) propose an alternative method for confidence calibration by analyzing internal representations. Detommaso et al. (2024) apply multicalibration, a strong and theoretically-grounded notion of calibration, to LLMs. These works typically rely on hypothesis testing or calibration-style assumptions on confidence estimates to control disagreement risk. In contrast, our work departs from assuming confidence monotonicity and instead treats it as a learnable property, formalized through a ranking-based formulation with PAC-Bayesian generalization guarantees. Related empirical analyses of LLM judge capability and domain-specific agreement, such as presupposition judgments (Atwell et al., 2025), further demonstrate that agreement varies significantly across tasks and metrics, motivating our focus on ranking generalization rather than raw correlation or average agreement. Similarly, critiques of simplistic agreement metrics in moral judgment settings (Grizzard et al., 2025) reinforce the need for stronger, distribution-aware guarantees, which our margin-based confidence ranking aims to provide.

There are other works study generalization and robustness of deep neural networks from the lens of PAC-Bayes (Jin et al., 2020; 2022a; 2025; 2022b; Yi et al., 2026). Other lines of work improve judge reliability through architectural or human-guided design choices rather than confidence learning (Zhou et al., 2025). Zhang et al. (2025) enhance LLM raters by modeling inferred reasoning traces, showing that richer internal representations can improve agreement with humans; our approach is complementary, as it operates at the level of confidence ordering and selective guarantees, independent of the internal reasoning mechanism of the judge. CheckEval (Lee et al., 2024) introduces a checklist-based, decomposed evaluation framework to improve robustness and interpretability, focusing on structured criteria rather than probabilistic guarantees. HREF (Lyu et al., 2024) leverages human responses to guide evaluation of instruction-following models, particularly relevant to datasets such as AlpacaEval and Chatbot Arena; unlike HREF, we do not rely on direct human-in-the-loop guidance at test time but instead learn a confidence estimator that generalizes beyond calibration data. Finally, work on metacognition and confidence divergence between humans and LLMs (Atwell et al., 2025) provides additional motivation for learning robust confidence estimators. Taken together, these approaches are largely complementary: while prior methods focus on richer judge architectures, task decomposition, or human guidance, our contribution centers on statistical generalization of confidence ranking as a principled route to reliable, scalable LLM evaluation with human-agreement guarantees.

# F. Additional Empirical Results

We provide more empirical results in this section.

*Table 5.* AUROC results of Mistral-7B under different values of $\beta$.

| Dataset | $\beta = 0$ | $\beta = 10^{-5}$ | $\beta = 10^{-4}$ | $\beta = 10^{-3}$ | $\beta = 10^{-2}$ |
|---|---|---|---|---|---|
| AlpacaEval | 0.6201 | 0.6401 | **0.6672** | 0.6643 | 0.6087 |
| HH-RLHF | 0.6159 | 0.6604 | **0.6805** | 0.6783 | 0.6290 |
| Chatbot Arena | 0.7034 | 0.7085 | 0.7127 | **0.7156** | 0.6961 |
| TL;DR | 0.6083 | 0.6347 | **0.6409** | 0.6372 | 0.6004 |

*Table 6.* Representative trajectory of $\gamma$ during training for Mistral-7B.

| Epoch | $\gamma$ **(Chatbot Arena)** | $\gamma$ **(HH-RLHF)** |
|---|---|---|
| 1 | 0.0109 | 0.0102 |
| 10 | 0.0420 | 0.0127 |
| 20 | 0.0564 | 0.0180 |
| 30 | 0.0626 | 0.0253 |

