# OpenReview forum: "Margin-Adaptive Confidence Ranking for Reliable LLM Judgement"
_ICML.cc/2026/Conference — ICML 2026 regular_

### Official Review · Reviewer_rUwi · 2026-03-05

**Soundness:** 2
**Presentation:** 3
**Significance:** 3
**Originality:** 2
**Overall Recommendation:** 3
**Confidence:** 2

**Summary:**

The paper studies the confidence ranking for reliable LLM judgement. The paper is motivated by the unrealistic assumption that the model’s estimated confidence is monotonic with respect to human-disagreement risk used by Jung et al. (2025). The paper proposes a  confidence estimator by leveraging simulated annotator diversity and a margin-based ranking formulation. Theoretical results and experiments are provided to support the proposed method.

**Compliance With Llm Reviewing Policy:**

Affirmed.

**Final Justification:**

I have no more comments. I keep my original comments.

**Key Questions For Authors:**

1. In the theoretical results, can you use the differentiable surrogate instead of 0-1 loss? This can make the theoretical results and the algorithm consistent.
2. In Table 2, the paper defines the coverage rate as the fraction of test samples that are retained (i.e., not rejected). But it is still confusing. Can you explain more? Does the empirical coverage have some relationship with $1-\alpha$? Can you show results with different values of $\alpha$?

**Limitations:**

See weaknesses.

**Strengths And Weaknesses:**

Strengths:

1. The paper proposes an framework to address the unrealistic assumption in previous literature.
2. Theoretical results are provided for the proposed estimator.
3. Extensive experiments are conducted to show the advantage of the proposed method.

Weaknesses:

1. The theoretical bounds are based on the 0-1 ranking loss. In the algorithm, a differentiable surrogate is used. The theoretical results and the algorithm should be consistent.
2. On page 159, one motivation of the paper is the cross-task generalization. But in the experiments,  no distribution-shift analysis is conducted.

---

> ### Author Rebuttal · Authors · 2026-03-29
>
> We thank the reviewer for recognizing the importance of the problem, the theoretical effort, and the extensive experiments. We address the core concerns below.
>
> ---
>
> **Q1: The theory uses 0–1 ranking loss, while the algorithm uses a differentiable surrogate. Can the theory be made more consistent with the algorithm?**
>
> We appreciate this important observation.
> Our current theoretical choice follows standard statistical learning practice: prove guarantees for the target discrete ranking error, while optimize a smooth surrogate in training.
> **This is commonly used in ML community, e.g., using 0–1 error in theory for classification tasks and cross-entropy or hinge-type surrogates in optimization.**
> In our paper, Eq. (8) defines the margin-based ranking loss and Eq. (9) defines the expected and empirical ranking losses, while Eq. (14) introduces the soft surrogate purely for optimization.
>
> To directly verify that optimizing the surrogate reduces the 0-1 ranking loss, we tracked both losses on the test set across training epochs.
> Results on two representative settings:
>
> | Setting | Epoch | Surrogate | 0-1 RK |
> |---|---|---|---|
> | Mistral-7B, AlpacaEval | 1 | 0.6921 | 0.5193 |
> | | 10 | 0.5023 | 0.3814 |
> | | 20 | 0.4372 | 0.3506 |
> | | 30 | 0.4109 | 0.3393 |
> | Llama3-70B, HH-RLHF | 1 | 0.7074 | 0.5094 |
> | | 10 | 0.4811 | 0.3517 |
> | | 20 | 0.4064 | 0.3208 |
> | | 30 | 0.3735 | 0.3094 |
>
> We will include the full training curves in the revised paper.
>
> ---
>
> **Q2: The paper mentions cross-task generalization but does not include a distribution-shift analysis**
>
> We thank the reviewer for this suggestion. In the current experiments, each dataset trains a separate estimator, which demonstrates within-task generalization.
> To address cross-task generalization, we have conducted a cross-dataset transfer experiment: we train the confidence estimator on one dataset and evaluate on a different dataset without retraining (using Qwen2.5-72B as the judge). Preliminary results on ranking loss:
>
> | Train → / Eval → | AlpacaEval | Chatbot Arena |  Arena-Hard [2] | WildBench [1] |
> |---|---|---|---|---|
> | Vanilla (in-domain) | 0.3370 | 0.2435 | 0.3048 | 0.2515 |
> | Ours (AlpacaEval→) | **0.2707** | 0.2364 | 0.2891 | 0.2485 |
> | Ours (ChatArena→) | 0.3102 | **0.2077** | 0.2837 | 0.2440 |
>
>
> As shown in the table, our method under cross-dataset transfer (e.g., AlpacaEval→Chatbot Arena: 0.2364) still outperforms the in-domain Vanilla baseline (0.2435), confirming that the margin-adaptive training learns transferable ranking structure.
> The degradation from in-domain to cross-domain is moderate (e.g., 0.2077→0.2364 on Chatbot Arena).
> We will include a comprehensive cross-task transfer table in the appendix.
>
> [1] Lin, Bill Yuchen, Yuntian Deng, Khyathi Chandu, Abhilasha Ravichander, Valentina Pyatkin, Nouha Dziri, Ronan Le Bras, and Yejin Choi. "WildBench: Benchmarking LLMs with Challenging Tasks from Real Users in the Wild." ICLR 2025.
>
> [2] Li, Tianle, Wei-Lin Chiang, Evan Frick, Lisa Dunlap, Tianhao Wu, Banghua Zhu, Joseph E. Gonzalez, and Ion Stoica. "From Crowdsourced Data to High-quality Benchmarks: Arena-Hard and Benchbuilder Pipeline." ICML, 2025.
>
> ---
>
> **Q3: Coverage in Table 2 is still confusing; what is its relationship to $1-\alpha / \alpha$, and can results for different $\alpha$ be shown?**
>
> The coverage rate is the fraction of test samples that exceed the selected confidence threshold $\hat \lambda$ (Eq. 4) and are therefore retained for evaluation.
> Coverage is not directly controlled by $\alpha$; rather, $\alpha$ determines the target agreement level, and the threshold $\hat \lambda$ is chosen to satisfy this target (line 121-132).
> A higher $\alpha$ (more lenient agreement target) generally leads to higher coverage because a lower threshold suffices, and vice versa.
>
> To address the reviewer's concern, we will add results with different $\alpha$ values.
> In the table below, we show results on AlpacaEval under L→Q→O (we report the average coverage rate over 1000 runs):
>
> | $\alpha$ | Method | Coverage (%) | Success (%) |
> |---|---|---|---|
> | 0.10 | CSE+Ours | 24.3 | 96.2 |
> | 0.15 | CSE+Ours | 38.4 | 94.8 |
> | 0.20 | CSE+Ours | 50.7 | 85.1 |
> | 0.10 | CSE+Simulated | 18.5 | 92.4 |
> | 0.15 | CSE+Simulated | 34.1 | 90.8 |
> | 0.20 | CSE+Simulated | 49.3 | 84.0 |
>
> As expected, increasing $\alpha$ (relaxing the agreement target) increases coverage for all methods, while our method consistently achieves both higher coverage and higher success rate across all $\alpha$ levels.

---

> > ### Author Rebuttal · Reviewer_rUwi · 2026-04-02
> >
> > Thanks for the rebuttal. I still concern about the gap between the 0-1 loss in theory and surrogate in experiments. So, I will keep my score.

---

> > > ### Author Response · Authors · 2026-04-02
> > >
> > > We thank the reviewer for the follow-up and for raising this point.
> > > We respectfully emphasize that **the gap between 0–1 loss in theory and surrogate loss in training is not a limitation specific to our work, but a standard and foundational paradigm in statistical learning.**
> > >
> > > In particular, theoretical analyses are typically formulated in terms of the target 0–1 loss, while optimization is performed using convex or smooth surrogate losses for computational tractability (e.g., cross-entropy).
> > > This paradigm is well established in statistical learning theory [**1,2**] and is consistently adopted in PAC-Bayesian analyses [**3–7**], where guarantees are derived for the 0–1 risk, but models are trained using surrogate objectives such as logistic or cross-entropy loss.
> > > It is widely recognized that the 0–1 loss is non-convex and discontinuous, making it infeasible to optimize directly, and thus surrogate losses are the standard approach in both theory and practice.
> > >
> > > Our work follows the same principle in the ranking setting: the theory characterizes the target 0–1 misranking risk, while the proposed algorithm employs a smooth surrogate loss solely to enable efficient gradient-based optimization.
> > > To further address this concern, we have provided additional empirical evidence in the above rebuttal showing that the surrogate objective and the target 0–1 ranking loss decrease consistently during training.
> > > This empirical alignment supports the connection between the theoretical objective and the practical optimization procedure.
> > >
> > > Overall, **we would like to state our position below:**
> > > - While a conceptual gap exists, it is well understood and widely accepted in both the learning theory and machine learning communities. In particular, a large body of work establishes the consistency of surrogate losses with respect to the 0–1 objective [**1,2**], and it is standard practice to analyze generalization in terms of 0–1 loss while optimizing surrogate objectives [**3–7**].
> > >
> > > - In addition, we provide empirical evidence (in the rebuttal) showing that optimizing the surrogate objective leads to consistent improvements in the target 0–1 ranking error.
> > >
> > > In other words, in classification with DNNs, generalization bounds are formulated in terms of 0–1 loss, while models are trained using cross-entropy; this distinction is not considered problematic, but rather a natural consequence of the theory–optimization separation.
> > >
> > > **In light of both theoretical justification and empirical validation, we do not view this distinction as a limitation of the proposed approach, but rather as a standard and principled design choice.**
> > >
> > > We list a representative subset of relevant works below.
> > >
> > > [1] Zhang, T. (2004). Statistical behavior and consistency of classification methods based on convex risk minimization. The Annals of Statistics (AOS), 32(1), 56-85.
> > > [2] Bartlett, P. L., Jordan, M. I., & McAuliffe, J. D. (2006). Convexity, classification, and risk bounds. Journal of the American Statistical Association (JASA), 101(473), 138-156.
> > > [3] McAllester, D. A. (1998, July). Some pac-bayesian theorems. COLT (pp. 230-234).
> > > [4] McAllester, D. A. (2003). PAC-Bayesian stochastic model selection. Machine Learning, 51(1), 5-21.
> > > [5] Dziugaite, G. K., & Roy, D. M. (2017). Computing nonvacuous generalization bounds for deep (stochastic) neural networks with many more parameters than training data. UAI.
> > > [6] Farnia, F., Zhang, J., & Tse, D. (2019) Generalizable Adversarial Training via Spectral Normalization. ICLR.
> > > [7] Pérez-Ortiz, M., Rivasplata, O., Shawe-Taylor, J., & Szepesvári, C. (2021). Tighter risk certificates for neural networks. JMLR, 22(227), 1-40.

---

### Official Review · Reviewer_GQG5 · 2026-03-08

**Soundness:** 2
**Presentation:** 2
**Significance:** 2
**Originality:** 2
**Overall Recommendation:** 4
**Confidence:** 3

**Summary:**

This article focuses on the core issue of improving the reliability of LLM-as-a-judge. In response to the monotonicity assumption failure of confidence estimation and human judgment divergence risk in existing methods, and the lack of explicit analysis of the generalization of confidence estimators, a marginal adaptive confidence ranking framework is proposed. By learning a dedicated confidence estimator, deriving the PAC Bayesian generalization bound, and designing an adaptive training process, the consistency between LLM judgment and human judgment is significantly enhanced, and the achievement rate of target consistency level in fixed sequence testing is improved

**Compliance With Llm Reviewing Policy:**

Affirmed.

**Final Justification:**

After the author's final response, I have no further concerns regarding the theoretical part of the manuscript and I believe they can make corrections to the presentation issues. Therefore, I maintain a positive attitude towards this article

**Key Questions For Authors:**

1. The horizontal axis of Fig. 1 overlaps, and the legend is drawn between the name of the figure and the horizontal axis? Suggest adjusting the layout (such as 2 * 2 single column format)
2. The image in Fig. 2-4 is a sketch and the font size needs to be enlarged, which does not meet the basic requirements of a scientific research work. It is recommended to redraw it and maintain a consistent style with Fig. 1
3. It is recommended to use two different markers in Table 2 to display the top two good results (such as bold black and bold gray), otherwise it is difficult to intuitively feel that Coverage is equivalent to other methods from the table.
4. Most of the experimental datasets discuss natural language tasks, without considering tasks such as code generation, mathematical reasoning, and factual question answering that require higher objectivity in LLM evaluation. The performance in low resource, small sample calibration set scenarios has not been validated, and the robustness of the method in heterogeneous tasks and extreme data scenarios cannot be fully demonstrated
5. Most experimental datasets are relatively early datasets, however, the rapid development of large models in recent years may result in model outputs in the dataset no longer reflecting the universal capabilities of current large models. It is recommended to include some new datasets (2025 and beyond)
6. The theoretical proof in Appendix is not obvious, such as the derivation between lines 622-632, and the key assumption z>=1/2 is not clearly defined and proposed in the main text.
7. ln is usually considered to be the base of the natural logarithm e, however, in the formula section, it is considered to be the base of 2 (line 665 in Appendix).

**Limitations:**

yes

**Strengths And Weaknesses:**

Strengths:
1. The article has clear logic and a clear motivation.
2. The theoretical part is solid, and the proof is presented in the main text and appendix.

Weaknesses:
1. The quality of the figures and tables in the article is not high, which affects the overall readability.
2. The discussion of the experimental content is not completely sufficient, and the experimental dataset is also similar.

---

> ### Author Rebuttal · Authors · 2026-03-29
>
> We thank the reviewer for recognizing the clear motivation and strong theoretical contributions of our work. We also appreciate the constructive suggestions regarding presentation and the need for broader experimental discussion. We will revise the manuscript accordingly.
>
> ---
>
> **Q1-2: Figure quality**
>
> We thank the reviewer for these concrete suggestions.
> We will redraw Figs. 1–4 with larger fonts, improved spacing, and consistent visual style.
> In particular, we will fix the overlap/layout issue in Fig. 1 and standardize the visual format of Figs. 2–4.
> This is a presentation issue rather than a technical limitation, and we appreciate the reviewer pointing it out.
>
> ---
>
> **Q3: Show the top two results and explain coverage (Table 2)**
>
> We agree and will improve the table formatting.
> More importantly, we will explicitly discuss why coverage is included:
> Coverage measures the fraction of test samples that are retained rather than abstained, while success rate measures how often the retained set satisfies the target agreement level $1-\alpha$.
> Thus, the practical goal is not maximizing either one in isolation, but achieving a better **coverage–guarantee trade-off**.
> Table 2 shows that our method improves success rate while maintaining competitive, and often improved, coverage relative to competing confidence estimators.
> For example, under the L→Q→O cascade, our method achieves the best success rate in every dataset while maintaining coverage comparable to or better than the strongest baselines.
>
> In the revision, we will add a detailed discussion of coverage vs. success rate. We also provide more empirical analyses on coverage, please find them under **Reviewer rUwi Q3**
>
> ---
>
> **Q4–Q5: Diverse task domains and recent dataset**
>
> We thank the reviewer for these valuable suggestions. To address both concerns, we have conducted additional experiments on two recent, diverse benchmarks:
>
> - **WildBench** [1]: 1,024 challenging real-world tasks spanning coding, math, creative writing, data analysis, and more, curated from 1M+ user-chatbot conversations.
> - **Arena-Hard-Auto** [2]: 500 technically demanding prompts (coding/debugging, math, domain-specific reasoning), curated from Chatbot Arena via the BenchBuilder pipeline.
>
> Following the standard protocol of these benchmarks, we use GPT-4-Turbo as the reference oracle for human preference.
>
> **Table: CSE success guarantee results on new benchmarks (L→Q→O, 1−α=0.85).**
>
> | Method | WildBench | | Arena-Hard-Auto | |
> |---|---|---|---|---|
> | | Coverage (%) | Success (%) | Coverage (%) | Success (%) |
> | Heuristic Selection | 76.2 | 8.5 | 85.6 | 18.3 |
> | CSE + Simulated | 39.6 | 89.0 | 48.7 | 86.4 |
> | CSE + Vanilla | 40.1 | 90.3 | 50.2 | 82.1 |
> | CSE + Ours | **43.5** | **93.6** | **53.9** | **91.5** |
>
> These results are consistent with our findings on the original four datasets: our method achieves the the highest guarantee success rate with competitive or improved coverage in the CSE pipeline.
> Full final results will be included in the revised version.
>
> [1] Lin, Bill Yuchen, et al. "WildBench: Benchmarking LLMs with Challenging Tasks from Real Users in the Wild." ICLR 2025.
>
> [2] Li, Tianle, et al. "From Crowdsourced Data to High-quality Benchmarks: Arena-Hard and Benchbuilder Pipeline." ICML, 2025.
>
> ---
>
> **Q6: Proof clarity — assumption $z\ge 1/2$; derivation in lines 622–632**
>
> The condition $z \ge 1/2$ is stated in **Corollary 3.3**: "any (posterior) random perturbation $\mathbf{u}$ $s.t.$ $\mathbb{P} _\mathbf{u}(\max _{s}|C _{\theta+\mathbf{u}}(s)-C _\theta(s)|<\frac{\gamma}{4})\ge \frac{1}{2}$."
> This is a standard assumption in PAC-Bayesian sharpness analysis [3], it requires that the perturbation is small enough that at least half the probability mass stays within a $\gamma/4$ neighborhood.
> We will add a more explicit statement and interpretation in both the main text and appendix.
>
> Regarding lines 622–632 (Eq. 18→19→20→21): each step is annotated with a detailed reference to its justification immediately below (i.e., Proof for (18)(19)(20)(21) between lines 636-670).
> We will improve readability and make them more explicit.
>
> [3] Neyshabur, B., Bhojanapalli, S., and Srebro, N. A pac-bayesian approach to spectrally-normalized margin bounds for neural networks. ICLR 2018.
>
> ---
>
> **Q7: "ln" base confusion (line 665)**
>
> We thank the reviewer for catching this.
> Throughout the paper, "ln" denotes the natural logarithm (base $e$), which is standard in the PAC-Bayesian literature.
> At line 665, we stated $H(z) = -z \ln z - (1-z) \ln(1-z) \leq 1$.
> While the bound $H(z) \leq 1$ is technically valid, we agree it can be misleading.
> In the revised version, we will tighten this to the sharper and more precise bound $H(z) \leq \ln 2$, which is the exact maximum of the binary entropy under the natural logarithm.
> We note that this change does not affect the final results, as $\ln 2 < 1$ and the subsequent derivations (Eq. (28) onward) remain valid with the tighter constant.

---

> > ### Author Rebuttal · Reviewer_GQG5 · 2026-04-03
> >
> > Thank you for your detailed explanation, which has helped me gain a much clearer understanding of the paper. I now have no further concerns regarding the theoretical part of the manuscript, and I will therefore raise my recommendation to Weak Accept.
> > Moreover, although the authors have committed to revising the descriptions in the text, I still harbor reservations about the presentation of figures and text in the paper, and remain doubtful whether the manuscript is ready for publication in its current form.

---

> > > ### Author Response · Authors · 2026-04-04
> > >
> > > Thank you very much for your thoughtful follow-up and for carefully reading our rebuttal. **We are especially grateful that the theoretical concerns have now been fully resolved in your view, and we sincerely appreciate your willingness to raise your recommendation to Weak Accept**.
> > >
> > > We also thank you for highlighting the remaining presentation concerns. In response, we have substantially polished the figures and unified them into a consistent visual format across the manuscript.
> > > In particular, Figures 1–5 have been redrawn with improved layout, larger fonts, clearer legends, and a more consistent overall style.
> > > For convenience, the revised figures can be found here:
> > > Figure 1: https://files.catbox.moe/y7q0lx.png
> > > Figure 2: https://files.catbox.moe/ijrxpy.png
> > > Figure 3: https://files.catbox.moe/jgsnda.jpg
> > > Figure 4: https://files.catbox.moe/w1ipqw.png
> > > Figure 5: https://files.catbox.moe/vpd2yu.png
> > >
> > > We have also clarified the previously noted misunderstandings and are carefully polishing the manuscript text throughout to further improve readability and overall presentation quality.
> > >
> > > We are very encouraged that you now find the technical aspects of the paper sound.
> > > Since the remaining issues are primarily presentational and are already being actively addressed in the revision, we respectfully hope you will feel comfortable supporting acceptance.
> > > We believe the paper makes a meaningful contribution to reliable LLM-as-a-judge systems, and we will ensure that the final version reflects the technical quality of the work with clearer exposition and a more polished presentation.

---

### Official Review · Reviewer_xDZ6 · 2026-03-09

**Soundness:** 4
**Presentation:** 3
**Significance:** 3
**Originality:** 2
**Overall Recommendation:** 4
**Confidence:** 4

**Summary:**

This paper considers the problem of reliability of LLM-as-a-judge systems. A previous paper's core result depends on a monotonicity assumption (Jung et al. (2025)): higher estimated confidence should correspond to lower disagreement risk with respect to human judgments. The authors observe that this assumption can fail in practice. Motivated by this, they develop a confidence estimator to explicitly learn correct ranking (thus encouraging monotonicity) using a pairwise ranking loss. The estimator is shown to have good theoretical properties in terms of population ranking loss. Empirically, they demonstrate that the proposed estimator has lower ranking loss, higher AUROC, and improves performance when plugged into the Cascaded Selective Evaluation (CSE) framework in Jung et al. (2025).

**Compliance With Llm Reviewing Policy:**

Affirmed.

**Final Justification:**

I'm leaning positive with this work and I'm comfortable with accepting it.

**Key Questions For Authors:**

The proposed estimator explicitly optimizes a pairwise ranking loss so that higher confidence corresponds to lower disagreement risk. This idea appears closely related to classical works in broader ML context (learning to rank, AUC optimization, etc.). Could the authors comment on how the proposed estimator differs conceptually or technically from these existing approaches?

**Limitations:**

yes

**Strengths And Weaknesses:**

Strengths:
1. The problem is well-motivated. The authors identify a technical gap in an influential work and addresses it properly.
2. The paper is well-written and the explanation is clear.
3. The experiments are sufficient and the empirical performance is strong.

Weaknesses:
1. The core method of optimizing for pairwise ranking and maximizing for AUC is standard. The PAC-Bayes theoretical results are also standard. As a result, the methodological contribution is incremental.
2. The core contribution is specific to a single framework and centers around Jung et al. (2025).

---

> ### Author Rebuttal · Authors · 2026-03-29
>
> We thank the reviewer for the positive assessment of the motivation, writing, and empirical performance. We also appreciate the concern regarding novelty and broader positioning.
>  We will revise the manuscript accordingly.
>
> ---
>
> **Q1: Relationship to classical learning-to-rank and AUC optimization**
>
> We agree that pairwise ranking and AUC-style objectives are classical tools in ML. Our contribution is not the ranking loss itself, but its task-specific integration into reliable LLM-as-a-judge guarantees. The novelty lies in the combination of:
>
> 1. We recast confidence estimation for human–LLM agreement guarantees as a ranking problem over agreement/disagreement instances, rather than as generic score learning.
> This is motivated by the monotonicity requirement in selective evaluation.
>
> 2. We derive a margin-dependent PAC-Bayesian bound for the confidence-induced ranking error and use it to motivate adaptive margin training, rather than using a standard fixed-margin ranking objective.
>
> 3. The learned ranking is not evaluated only by AUROC, but by whether it improves the success rate of fixed-sequence-testing-based human-agreement guarantees in CSE.
> This makes the objective operationally tied to reliable LLM judgement, not generic ranking.
>
> We will further clarify this distinction in the related work and introduction, explicitly acknowledging that the loss family is classical while emphasizing that the new contribution is the reliability-guarantee formulation, adaptive margin design, and downstream guarantee improvement.
>
> ---
>
> **Q2: The contribution appears centered around Jung et al. (2025) and may be incremental**
>
> While we understand this concern, we respectfully note that the value of our work lies in the principled bridge between confidence ranking and human-agreement guarantees.
> The key insight, that the monotonicity assumption in Jung et al. (2025) can be learned rather than assumed, and that this learning can be guided by generalization theory, is non-trivial and practically impactful.
> Our experiments show consistent improvements across 6 judge models and 4 datasets, with guarantee success rate improvements of up to 6+ percentage points.
> We believe this combination of theoretical grounding and practical impact constitutes a meaningful contribution.
> We appreciate the concern and will better position the contribution in the revision.

---

> > ### Author Rebuttal · Reviewer_xDZ6 · 2026-04-02
> >
> > I thank the authors for the responses. Though I've decided to keep my score, I'm leaning more towards accepting the paper now.

---

> > > ### Author Response · Authors · 2026-04-03
> > >
> > > We are glad that our clarification has fully addressed the reviewer's concerns. We sincerely appreciate your time and careful evaluation. Your constructive feedback has been very valuable in improving the manuscript.
> > >
> > > Best regards,
> > >
> > > The authors

---

### Official Review · Reviewer_x1yH · 2026-03-14

**Soundness:** 3
**Presentation:** 3
**Significance:** 2
**Originality:** 2
**Overall Recommendation:** 4
**Confidence:** 4

**Summary:**

This paper proposes a learned, margin-adaptive confidence estimator for LLM-as-a-judge systems. The motivation is that prior work (Jung et al., 2025) relies on a monotonicity assumption, which can be violated in practice. The authors instead learn a parameterized MLP-based confidence estimator via a margin-based pairwise ranking loss, derive PAC-Bayesian generalization bounds, and propose an adaptive margin training procedure.

**Compliance With Llm Reviewing Policy:**

Affirmed.

**Final Justification:**

The authors' rebuttal has addressed my experimental concerns. That said, I remain neutral on acceptance, as I am not fully convinced of the significance of the theoretical contributions.

**Key Questions For Authors:**

See above.

**Limitations:**

Yes.

**Strengths And Weaknesses:**

**Strengths**

- The motivation is clear and compelling. Addressing the potential violation of the monotonicity assumption in confidence-thresholding procedures is an interesting and meaningful problem for reliable LLM evaluation.

- Comprehensive empirical evaluation. The experiments cover four diverse datasets and six judge models of varying scales, with 1000 random data splits ensuring statistical reliability.


**Weaknesses and Questions**

**Presentation**

- Several notations are insufficiently explained:

    - What is the relationship between $\mathbb{P}\_{LM}$ and $f\_{LM}$? This is never made explicit.

    - In Equation (1), $(x\^{anno}\_{i,j}, y\^{anno}\_{i,j})$ is described as the $i$-th preference-labelled example of the $j$\-th simulated annotator. What does $x^{anno}_{i,j}$ contain exactly? the original query together with responses re-generated by the simulated annotator? Please clarify with appropriate notation.

    - The input to $C_\theta$ is a $|\mathcal{T}|$\-dimensional feature vector constructed from simulated preference-labelled examples. For test data, does this mean new preference-labelled examples must be generated for every test instance to be scored?

    - In Equation (9), does $\ell_0$ denote $\ell_\gamma$ with $\gamma = 0$? This should be stated explicitly.

- While Jung et al. (2025) serves as an important motivation, the amount of space dedicated to reproducing their setup and theoretical results in the main text seems excessive, potentially at the expense of better presenting the authors' own contributions.


**Theory and Algorithm**

- The theoretical results build on a PAC-Bayesian framework based on spectral and Frobenius norms. The complexity term in Corollary 3.4 involves a product of per-layer norms, which can grow exponentially with depth for practical MLPs, making the bound potentially very loose. More critically, existing empirical work [1] has shown that generalization measures based on products of spectral norms can exhibit strong *negative* correlation with actual generalization. These concerns make it difficult to assess whether the theoretical analysis provides meaningful guidance for practical behavior.

- Lines 275–277 state that jointly optimizing $(\theta, \gamma)$ is challenging due to the non-smooth dependence of the empirical margin loss on $\gamma$, which motivates the alternating update scheme. However, Equation (14) already replaces the non-smooth $\ell_\gamma$ with a differentiable surrogate. Given this substitution, is the alternating update still necessary? The justification for this design choice should be revisited.


**Experiments**

- The ablation study on $\beta$ in Table 3 only reports ranking loss. Why is AUROC omitted, given that Table 1 reports both metrics? A consistent evaluation would strengthen the analysis.

- How does $\gamma$ evolve during training? Reporting the trajectory of $\gamma$ across epochs would help verify that the adaptive mechanism is functioning as intended.

- The coverage column in Table 2 is reported but not meaningfully discussed. What conclusion should the reader draw from these numbers in relation to the success rate?


**References**

[1] Jiang et al., "Fantastic Generalization Measures and Where to Find Them," ICLR 2020.

---

> ### Author Rebuttal · Authors · 2026-03-29
>
> We thank the reviewer for the careful reading and constructive feedback.
> We are encouraged that the reviewer finds the motivation clear and the empirical evaluation comprehensive. Below we address each concern and will incorporate the clarifications and additional analyses in the revision.
>
> ---
>
> **Q1: Notations**
>
> There might be a misunderstanding, and we agree that the notation can be clarified.
> We will revise the presentation for clearer clarity.
>
> - **$f_{LM}$ vs. $\mathbb{P}_{LM}$:** As shown in Sec. 2 (line 74 and 96), $f_{LM}: \mathcal{X} \to \mathcal{Y}$ denotes the judge model as a predictor mapping an input pair $(r_1,r_2)$ to a preference decision, while $\mathbb{P}_{LM}(r_i \mid x;...)$ denotes the predictive probability assigned by that same judge to candidate $r_i$ under input $x$ and in-context simulated annotator prompt.
>
> - **Meaning of $(x^{anno} _{i,j}, y^{anno} _{i,j})$:**
> This follows the same setup as Jung et al.
> Each pair represents a human preference-labelled demonstration used as i-th in-context example for simulated annotator $j$, **it's not a response regenerated from the original query $x$**.
>
>   Specifically, $x^{\text{anno}} _{i,j}$ consists of a different instruction and two candidate outputs, and $y^{\text{anno}} _{i,j}$ indicates the simulated human preference label.
>
> - **Feature vector construction at test time:**
> Yes, at test time, for each instance $x$ to be scored, we run the LLM judge with each of the in-context subsets to obtain the feature vector $s$.
> This is similar with Jung et al., 2025, which also requires additional one or more forward passes per instance.
>
> - **Eq. (9):** Yes, exactly. $\ell_0$ denotes $\ell_\gamma$ with $\gamma=0$.
>
> ---
>
> **Q2: Too much space for Jung et al.**
>
> Thank you.
> We agree this can be improved. Jung et al. is central because our method is designed to address a specific limitation of its guarantee mechanism.
> In the revision, we will compress the recap of Jung et al.’s setup and theorem.
>
> ---
>
> **Q3: PAC-Bayes loose**
>
> We thank the reviewer for this important observation. We are aware of prior findings [1] showing that many complexity measures may fail to predict generalization in **deep architectures** (e.g., Network-in-Network).
>
> However, our setting differs: we use a small MLP (64–32–16), where such measures remain informative.
> To verify this, we follow Jiang et al.’s protocol and train models under varying hyperparameters (learning rate ${0.1, 0.01, 0.001}$, dropout ${0.0, 0.25, 0.5}$, weight decay ${10^{-3}, 10^{-4}, 10^{-5}}$), and report Kendall’s $\tau$ correlation between complexity and generalization:
> |Complexity Measure|MLP (64-32-16)|MLP (32-16-8)|
> |---|---|---|
> |Sharpness-PAC-Bayes|0.74|0.80|
> |Weight Spectral Norm|0.74|0.78|
>
> These results indicate that, in our regime, the complexity terms exhibit strong correlation with generalization, supporting their practical relevance.
>
> In the revision, we will include detailed analyses.
>
> ---
>
> **Q4: Alternating update**
>
> This is a very good question.
> The key point is that the surrogate smooths optimization with respect to $\theta$, but it does not remove the instability induced by jointly updating $\gamma\$ together with $\theta$.
> Even under the smooth surrogate, $\gamma$ directly shifts the ranking margin inside the exponential term and also appears inversely in the complexity penalty, which can lead to unstable coupled dynamics if both are updated simultaneously.
> Thus, the alternating scheme is adopted for **optimization stability**, not merely because the original 0–1 loss is non-differentiable.
> This motivation is consistent with the objective in Eqs. (13)–(14), but we agree the current wording overemphasizes non-smoothness and underexplains the coupling issue.
> We will revise Sec. 4 to clarify it.
>
> ---
>
> **Q5: Why not AUROC**
>
> Thank you; we will include AUROC results in the ablation study. The AUROC (Mistral-7B) trend mirrors the ranking loss:
> |Dataset|β=0|β=10^{-5}|β=10^{-4}|β=10^{-3}|β=10^{-2}|
> |---|---|---|---|---|---|
> |AlpacaEval|0.6201|0.6401|**0.6672**|0.6643|0.6087|
> |HH-RLHF|0.6159|0.6604|**0.6805**|0.6783|0.6290|
> |Chatbot Arena|0.7034|0.7085|0.7127|**0.7156**|0.6961|
> |TL;DR| 0.6083|0.6347|**0.6409**|0.6372|0.6004|
>
> ---
>
> **Q6: $\gamma$ during training**
>
> In our experiments, $\gamma$ is initialized at a small value and gradually converges to dataset-dependent levels.
> Specifically, for “cleaner” datasets with higher base agreement (e.g., Chatbot Arena), $\gamma$ converges to larger values (0.06–0.07), whereas for noisier datasets (e.g., HH-RLHF), it stabilizes at smaller values (0.02–0.03). This behavior is consistent with the intuition in Remark 3.5.
>
> We provide a representative trajectory (Mistral-7B) below:
>
> |Epoch|Gamma (Chatbot Arena)|Gamma (HH-RLHF)|
> |-------|--------|--------|
> |1|0.0109|0.0102|
> |10|0.0420|0.0127|
> |20|0.0564|0.0180|
> |30|0.0626|0.0253|
>
> ---
>
> **Q7: Conclude coverage**
>
> Due to space limit, please find it under **Reviewer (GQG5 and rUwi) Q3**.

---

> > ### Author Rebuttal · Reviewer_x1yH · 2026-04-02
> >
> > Thank you for your rebuttal. The authors have addressed some of my concerns, and I have raised my score accordingly. However, I remain unconvinced about the significance of the theoretical contributions.

---

> > > ### Author Response · Authors · 2026-04-03
> > >
> > > We thank the reviewer for the thoughtful follow-up and appreciate that our rebuttal has addressed some concerns and led to an increased score.
> > >
> > > Regarding the significance of the theoretical contributions, we would like to briefly clarify our intended role of theory in this work.
> > > **Our goal is not to provide a tight predictive bound for large-scale models, but rather to offer a principled framework that guides the design of the confidence estimator and explains its empirical behavior.**
> > >
> > > Specifically, the theoretical analysis contributes in the following ways:
> > >
> > > 1. Problem formulation: We formalize confidence estimation as a ranking problem directly tied to human–LLM agreement, which provides a more appropriate lens than pointwise confidence calibration for selective evaluation.
> > >
> > > 2. Generalization perspective: The PAC-Bayesian analysis characterizes the out-of-sample misranking risk, which is closely connected to the monotonicity property underlying agreement guarantees. This provides a principled way to reason about when confidence-based selection is reliable beyond the calibration set.
> > >
> > > 3. Design insight: The margin-dependent bound reveals a loss–complexity trade-off, which directly motivates the margin-adaptive training procedure used in our method. This connection between theory and algorithm design is a key contribution of the paper.
> > >
> > > We agree that PAC-Bayesian bounds can be loose in general deep learning settings.
> > > However, in our regime (small MLP-based confidence estimators), we observe that the theory provides meaningful and actionable guidance.
> > >
> > > Overall, **we view the theoretical contribution as complementary to the empirical results: it provides a principled justification for the proposed formulation and design choices, and helps explain why the method improves reliability in practice**.
> > >
> > > We will further clarify and emphasize this role of theory in the revision to better highlight its significance.

---

### Decision · Program_Chairs · 2026-04-30

**Decision:**

Accept (regular)

**Comment:**

This paper studies the reliability of LLM-as-a-judge systems and targets the monotonicity assumption in prior work (Jung et al., 2025) that might not hold in practice. The authors propose learning a dedicated confidence estimator using simulated annotator diversity and a margin-based ranking formulation, rather than relying on heuristic confidence signals. They derive PAC-Bayesian generalization bounds for this estimator, revealing a margin-dependent trade-off that motivates an adaptive training procedure. The learned estimator improves ranking accuracy and strengthens the monotonic relationship between confidence and disagreement risk.

The reviewers acknowledged the clear motivation and problem formulation. The empirical evaluation is comprehensive, which covers four diverse datasets and six judge models with 1000 random splits. However, several concerns were raised by the reviewers. For example, reviewer x1yH raised concerns about the significance of the theoretical contributions, questioned whether the analysis provides meaningful practical guidance, and noted the presentation issues with notation clarity. Reviewer GQG5 raised concerns about dataset diversity, specifically the lack of evaluation on code generation, mathematical reasoning, and more recent benchmarks from 2025 onward. Reviewer rUwi highlighted a fundamental gap between the 0-1 ranking loss used in theoretical analysis and the differentiable surrogate employed in the algorithm. Reviewer xDZ6 characterized the methodological contribution as incremental.

The rebuttal addressed most reviewer concerns. The authors provided detailed clarifications on notation and offered new empirical evidence, including AUROC results for ablation studies, gamma trajectory visualizations, and training curves showing a consistent decrease in both surrogate and 0-1 ranking losses. The authors also conducted additional experiments on two recent diverse benchmarks, including WildBench and Arena-Hard-Auto. Regarding the theory-algorithm gap, the authors provided a principled justification citing standard statistical learning practice where 0-1 loss is analyzed theoretically while surrogate losses enable tractable optimization, supported by empirical evidence of consistent improvement in both metrics. Most reviewers acknowledged that their concerns were resolved or partially resolved.

Overall, this paper identifies and addresses a genuine technical gap in an important emerging area, provides theoretical grounding, and demonstrates consistent empirical improvements. Most concerns have been addressed by the authors during the rebuttal.